# Absence of HTATIP2 Expression in A549 Lung Adenocarcinoma Cells Promotes Tumor Plasticity in Response to Hypoxic Stress

**DOI:** 10.3390/cancers12061538

**Published:** 2020-06-11

**Authors:** Minghua Li, Jing Li, Xiaofang Guo, Hua Pan, Qingyu Zhou

**Affiliations:** 1Department of Pharmaceutical Sciences, Taneja College of Pharmacy, University of South Florida, Tampa, FL 33612, USA; minghua@usf.edu (M.L.); xiaofang@usf.edu (X.G.); 2Karmanos Cancer Institute, Wayne State University School of Medicine, Detroit, MI 48201, USA; LiJing@wayne.edu; 3Division of Cardiovascular Sciences, Morsani College of Medicine, University of South Florida, Tampa, FL 33612, USA; huapan@usf.edu

**Keywords:** tumor hypoxia, HTATIP2, hypoxia inducible factors, epithelial-mesenchymal transition, tumor metabolic plasticity

## Abstract

HIV-1 Tat Interactive Protein 2 (HTATIP2) is a tumor suppressor, of which reduced or absent expression is associated with increased susceptibility to tumorigenesis and enhanced tumor invasion and metastasis. However, whether the absent expression of HTATIP2 is a tumor-promoting factor that acts through improving tumor adaptation to hypoxia is unclear. Here, we established a stable HTATIP2-knockdown A549 human lung adenocarcinoma cell line (A549shHTATIP2) using lentiviral-delivered HTATIP2-targeting short hairpin RNA (shRNA), employed a double subcutaneous xenograft model and incorporated photoacoustic imaging and metabolomics approaches to elucidate the impact of the absent HTATIP2 expression on tumor response to hypoxic stress. Results from the *in vivo* study showed that A549shHTATIP2 tumors exhibited accelerated growth but decreased intratumoral oxygenation and angiogenesis and reduced sensitivity to sorafenib treatment as compared with their parental counterparts. Moreover, results of the immunoblot and real-time PCR analyses revealed that the HIF2α protein and mRNA levels in vehicle-treated A549shHTATIP2 tumors were significantly increased (*p* < 0.01 compared with the parental control tumors). Despite the strong HIF2α-c-Myc protein interaction indicated by our co-immunoprecipitation data, the increase in the c-Myc protein and mRNA levels was not significant in the A549shHTATIP2 tumors. Nonetheless, MCL-1 and β-catenin protein levels in A549shHTATIP2 tumors were significantly increased (*p* < 0.05 compared with the parental control tumors), suggesting an enhanced β-catenin/c-Myc/MCL-1 pathway in the absence of HTATIP2 expression. The finding of significantly decreased E-cadherin (*p* < 0.01 compared with vehicle-treated A549shHTATIP2 tumors) and increased vimentin (*p* < 0.05 compared with sorafenib-treated A549 tumors) protein levels in A549shHTATIP2 tumors implicates that the absence of HTATIP2 expression increases the susceptibility of A549 tumors to sorafenib-activated epithelial-mesenchymal transition (EMT) process. Comparison of the metabolomic profiles between A549 and A549shHTATIP2 tumors demonstrated that the absence of HTATIP2 expression resulted in increased tumor metabolic plasticity that enabled tumor cells to exploit alternative metabolic pathways for survival and proliferation rather than relying on glutamine and fatty acids as a carbon source to replenish TCA cycle intermediates. Our data suggest a mechanism by which the absent HTATIP2 expression modulates tumor adaptation to hypoxia and promotes an aggressive tumor phenotype by enhancing the HIF2α-regulated β-catenin/c-Myc/MCL-1 signaling, increasing the susceptibility of tumors to sorafenib treatment-activated EMT process, and improving tumor metabolic plasticity.

## 1. Introduction

Hypoxia is a central feature of malignant tumors and affects many other hallmarks of cancer, including promotion of tumor cell survival and proliferation, resistance to apoptosis, a switch to anaerobic metabolism, increase in angiogenesis, and activation of invasive growth [1]. Cellular adaptation to hypoxia is primarily mediated by a family of transcription factors termed hypoxia inducible factors (HIFs). HIF is a heterodimeric protein complex consisting of two members of the basic helix-loop-helix Per-ARNT-Sim (bHLH-PAS) family of transcription factors, an oxygen dependent α subunit and a constitutively expressed β unit [2]. The HIF signaling is mainly regulated by the stability of two HIF α subunit transcription factors, HIF1α and HIF2α. Under hypoxic conditions, the HIF α subunits induce regulatory cascades that rewire oncogenic signaling pathways and tumor metabolism, leading to enhanced aggressiveness of tumors and reduced tumor response to therapeutic intervention [3].

HIF1α and HIF2α display differential tissue distribution and regulate HIF target genes in diverse biological pathways [3]. HIF1α is ubiquitously present in most cell types, while HIF2α is selectively expressed in certain tissues, such as lungs, heart, retina and glial cells [4,5]. HIF1α and HIF2α share 48% amino acid sequence homology with similar structures of their DNA binding and dimerization domains, but with different transactivation domains, thereby regulating different sets of target genes [6]. It has been demonstrated that HIF1α is activated for a short period of time (2–24 h) under severe hypoxia (<0.1% O_2_), whereas HIF2α was activated for an extended period of time (48–72 h) under mild hypoxia (<5% O_2_) [7,8]. The different time course of induction by hypoxia implies that switching between HIF1α and HIF2α dependency mobilizes cellular adaptation to hypoxia in tumors [9]. Studies on the transcriptional selectivity of HIF α subunits have shown that HIF1α and HIF2α have little functional redundancy but play complementary roles in regulating cellular transcriptional response to hypoxia [7,10,11,12]. A recent study on the transcriptional outputs of stabilized forms of HIF1α and HIF2α revealed that HIF1α and HIF2α regulated 701 and 1454 genes, respectively. The HIF1α-regulated genes are primarily involved in regulation of metabolic reprogramming, while the HIF2α target genes regulate angiogenic extracellular signaling, guidance cues, and extracellular matrix remodeling factors. HIF2α exclusively regulated distinct subsets of transcription factors and co-regulators pertaining to its own multifaceted roles in hypoxia [11]. It is evident that tumor adaptation to hypoxia is regulated by a balance between HIF1α and HIF2α, and alterations in the balance steer the HIF-regulated tumor-promoting processes.

HTATIP2, also known as 30-kDa HIV-1 Tat interacting protein (TIP30) or CC3, is characterized as a tumor suppressor, and downregulation of HTATIP2 has been associated with poor prognosis in patients with different types of cancer, including hepatocellular carcinoma (HCC) [13], gastric cancer [14], pancreatic cancer [15], breast cancer [16], colon cancer [17], glioblastoma [18], and lung cancer [19,20]. A good amount of evidence has indicated that the pro-apoptotic and anti-metastatic activities of HTATIP2 are associated with its modulatory effect on the expression of certain pro-apoptotic (e.g., Bad, Bax, Siva, p53), anti-apoptotic (e.g., Bcl-xl, MCL-1) and metastasis suppressor genes (e.g., NM23-H2) through phosphorylation of the heptapeptide repeats of the C-terminal domain of the largest subunit of DNA-dependent RNA polymerase II [21,22,23,24]. Moreover, a study on the role of HTATIP2 in cancer metabolism demonstrated that silencing of HTATIP2 in HeLa cells cultured under low glucose condition improved their metabolic adaptation to glucose limitation by maintaining high levels of c-Myc and the M2 isoform of pyruvate kinase [25]. In addition to being a marker of poor prognosis in a variety of human cancer, HTATIP2 downregulation was found to be associated with the acquired drug resistance. A recent study showed that HTATIP2 expression was decreased significantly in tumor cells derived from the Hep2 laryngeal squamous cell carcinoma (LSCC) xenograft tumors that were carried by BALB/c nude mice treated with cisplatin once a week for 5 weeks [26]. Those drug-selected xenograft-derived tumor cells were resistant not only to cisplatin, but also to 5-flourouracil [26]. A series of preclinical and clinical studies on the acquired resistance to sorafenib, a multikinase inhibitor targeting the serine/threonine kinase *RAF* as well as the VEGF and PDGF *receptor tyrosine kinases* [27], have demonstrated that downregulation of HTATIP2 by sorafenib in HCC was associated with the activated epithelial-mesenchymal transition (EMT) process and increased invasive and metastatic potentials [28,29,30]. Despite the evidence that decreased HTATIP2 expression is indicative of the development of cancer chemoresistance, the exact mechanism remains unclear. Additional recent studies reported that metformin enhanced the anti-tumor effect of sorafenib [31] or regorafenib [32] by reducing HIF2α expression and increasing HTATIP2 expression at the protein level. Although the results of those studies implicate HTATIP2 as a HIF2α target gene involved in cancer progression and chemoresistance, questions remain about whether HTATIP2 is related to another key modulator of cellular response to hypoxia stress, HIF1α, and whether repression of HTATIP2 contributes to tumor progression by fine-tuning the balance between HIF1α and HIF2α given the fact that HIF1α and HIF2α have distinct but complementary roles in hypoxic adaptation and exhibit antagonistic activities at times [33].

In the present study, we elucidated a novel mechanism underpinning the impact of the absence of HTATIP2 expression on the activation of HIF signaling that mediates tumor adaptation to hypoxia and subsequently promotes aggressive tumor growth and resistance to therapy in a murine xenograft model of A549 human lung adenocarcinoma, which represents the most common subtype of non-small cell lung carcinoma (NSCLC).

## 2. Results

### 2.1. Knockdown of HTATIP2 in A549 Cells Affected Cell Migration but Had Little Impact on Invasion and Response to Sorafenib Treatment in Vitro under Normoxic and Hypoxic Conditions

To determine whether the absence of HTATIP2 expression modulates tumor cell adaptation to hypoxia, stable non-target (A549shNT) and HTATIP2-knockdown (A549shHTATIP2) A549 cell lines were generated by transducing A549 cells with a non-target shRNA and a HTATIP2-specific shRNA, respectively, through lentiviral infection. Comparison of the doubling time values showed no significant difference between A549shNT and A549shHTATIP2 cell lines (*n* = 4; 19.2 ± 1.2 h versus 20.2 ± 2.4 h, *n* = 4, *p* > 0.05). Next the *in vitro* migration and invasion potentials of A549shNT and A549shHTATIP2 cells were assessed under normoxic and hypoxic conditions using the wound-healing assay and transwell invasion assay. Results of the wound-healing assay indicated that the mean percent wound closure values of A549shNT cells were significantly lower than that of A549shHTATIP2 cells at 48 h after wounding under both normoxic (16% decrease) and hypoxic (26% decrease) conditions (*n* = 8; *p* < 0.01 for both conditions) (Figure 1A,B), indicating that A549shHTATIP2 cells possess greater migration potential than A549shNT cells under the same culture condition. Results of the 14-h transwell assay showed no significant difference in the invasion potential between A549shNT and A549shHTATIP2 cells under normoxic (27% ± 9% vs. 18% ± 3%) and hypoxic (28% ± 5% vs. 28% ± 6%) conditions (*n* = 3; *p* > 0.05 for both conditions) (Figure 1C,D). Since downregulation of HTATIP2 has been associated with acquired sorafenib resistance [28,29,30], a cytotoxicity study was carried out to examine the impact of HTATIP2 knockdown on tumor cell response to sorafenib treatment *in vitro* under normoxic and hypoxic conditions. Results of the quantitative analysis of the sorafenib dose-effect relationship demonstrated that there was no significant difference in sorafenib IC50 values between A549shNT and A549shHTATIP2 cells under the same culture condition, suggesting that HTATIP2 knockdown does not affect the cytotoxicity of sorafenib *in vitro*. For both A549shNT and A549shHTATIP2 cells, the sorafenib IC50 values obtained under hypoxic condition were significantly greater than those obtained under normoxic conditions (*n* = 8; *p* < 0.01 for both cell lines) (Figure 1E), suggesting that hypoxia promotes tumor cell resistance to sorafenib treatment *in vitro* regardless of the presence or absence of HTATIP2 expression. All of these results illustrate that knockdown of HTATIP2 increases the migratory potential of A549 cells with no significant effect on cell proliferation, invasion and response to sorafenib treatment *in vitro*.

### 2.2. Examination of Molecular Changes in Response to Hypoxic Stress and Sorafenib Treatment in Cultured A549shNT and A549shHTATIP2 Cells

It is known that HIF1α and HIF2α are key regulators of cellular response to hypoxia [3], and activation of PI3K/Akt pathway is associated with sorafenib resistance [34]. To explore the impact of HTATIP2 knockdown on the HIF-mediated tumor cell response to hypoxic stress and therapeutic insult, the baseline and treatment-induced changes in the expression levels of selected proteins in cultured A549shNT and A549shHTATIP2 cells were examined (Figure 2A,B and Appendix A. Those proteins included HIF1α, HIF2α and their target genes c-Myc and β-catenin, as well as MXI1 and MCL-1 that are closely associated with c-Myc, EMT markers and effectors of the Akt signaling pathway that are potentially associated with tumor response to sorafenib treatment. As shown in Figure 2A,B, in the absence of sorafenib treatment, both HIF1α and HIF2α were highly expressed in both A549shNT and A549shHTATIP2 cell lines that were exposed to hypoxia (0.5% O_2_) for 24 h. HIF1α expression was undetectable and HIF2α expression was barely detectable in both cell lines cultured under normoxic condition. Of all the proteins assessed, only phospho-Akt level was significantly changed (decreased by 75%) in the hypoxic A549shHTATIP2 cells as compared with the hypoxic A549shNT cells (*n* = 3; *p* < 0.01) (Figure 2A,B). Sorafenib treatment appeared to decrease the expression of HIF1α, HIF2α, c-Myc, MXI1 and MCL-1, and increase the expression of vimentin and phospho-Akt in both A549shNT and A549shHTATIP2 cells under normoxia and hypoxia conditions. Notably, the decrease in MXI1 expression was not statistically significant in A549shHTATIP2 cells regardless of the culture condition, nor was the increase in vimentin expression in hypoxic A549shHTATIP2 cells (Figure 2A,B). Moreover, under hypoxic condition, in the presence of sorafenib, the expression levels of phospho-Akt (*p* < 0.01) and phospho-mTOR (*p* < 0.05) in A549shHTATIP2 cells were significantly lower than those in the A549shNT cells, whereas the expression levels of HIF2α and MXI1 in the A549shHTATIP2 cells were significantly higher than those in the A549shNT cells (*p* < 0.05 for both proteins). These findings implicate that HTATIP2 knockdown potentially affects certain signaling pathways involved in tumor cell response to sorafenib treatment under hypoxic condition. In light of the observation that sorafenib reduced HIF2α and MXI1 protein levels to a lesser extent in the hypoxic A549shHTATIP2 cells as compared to the A549shNT cells and given the fact that hypoxic regulation of MXI1, a c-Myc antagonist, is HIF1-dependent [35], the possible interaction of HTATIP2 with HIF1α, HIF2α and c-Myc in A549 cell lysates using co-immunoprecipitation was examined. Results of the co-immunoprecipitation assay using the whole cell extracts from A549 cells cultured under hypoxia condition showed that HIF1α and HTATIP2 co-immunoprecipitated with HIF2α and so did HIF2α co-immunoprecipitate with antibodies recognizing HIF1α or HTATIP2. HIF1α was not detected in the immunoprecipitates of HTATIP2 and neither was HTATIP2 detected in the immunoprecipitates of HIF1α (Figure 2C and Appendix A). The c-Myc co-immunoprecipitation was seen with all immunoprecipitates. In particular, a large amount of c-Myc was associated with the HIF2α immunoprecipitated complexes, reflecting a strong protein-protein interaction between HIF2α and c-Myc (Figure 2C). Overall the co-immunoprecipitation presented in Figure 2C suggests that HTATIP2 interacts with both HIF2α and c-Myc but not with HIF1α.

### 2.3. Absence of HTATIP2 Expression Promoted Tumor Growth, Decreased Tumor Oxygenation and Reduced Sensitivity to Sorafenib Treatment without Affecting Sorafenib Tumor Distribution

Since results from the *in vitro* study produced a hint that the modulatory effect of HTATIP2 knockdown on the *in vitro* behavior of A549 cells might be associated with HIF2α-mediated tumor adaptation to hypoxia, a subcutaneous tumor xenograft model was employed to further investigate the effect of the absent HTATIP2 expression on the phenotypic behavior of growing tumors. Following the subcutaneous inoculation of the same number of parental A549 and A549shHTATIP2 cells in opposite flanks of the same animal, the growth of A549shHTATIP2 tumor xenografts appeared to be faster than that of A549 xenografts irrespective of the presence or absence of sorafenib treatment (Figure 3A). Results of the fold change of tumor volume, which was calculated as ratio of tumor volume on Day 5 or Day 10 of the vehicle or sorafenib treatment to tumor volume on Day 0 (before the start of the treatment), demonstrated that the fold change of volume of the sorafenib-treated A549 tumors on Day 5 of the treatment was significantly lower than that of the vehicle-treated A549 tumors (1.06 ± 0.04 versus 1.16 ± 0.04, *p* < 0.01), and that of the sorafenib-treated A549shHTATIP2 tumors (versus 1.20 ± 0.15, *p* < 0.05) (Figure 3B). The fold change of tumor volume on Day 5 of the treatment was not significantly different between vehicle- and sorafenib-treated A549shHTATIP2 tumors (1.34 ± 0.23 versus 1.20 ± 0.15, *p* > 0.05). No significant difference in the fold change in tumor volume on Day 10 of the treatment was found between different groups (*p* > 0.05 using the two-sample *t* test) (Figure 3B). Consistent with the fold change of tumor volume data, the mean weight of vehicle control A549shHTATIP2 xenograft tumors was significantly greater than that of their parental counterparts (84.7 ± 32.3 mg versus 26.3 ± 5.4 mg; *p* < 0.01) (Figure 3C,D). The difference in weights between sorafenib-treated A549shHTATIP2 and A549 tumors was not statistically significant, which was in part due to large inter-individual variability (83.5 ± 77.9 mg versus 19.5 ± 5.9 mg, *p* = 0.072) (Figure 3C,D). Collectively, the tumor growth data indicated that the A549shHTATIP2 tumors exhibited accelerated growth and reduced sensitivity to sorafenib treatment as compared with the A549 tumors. With regard to the intratumoral oxygenation measured before the start of the vehicle and sorafenib treatment (denoted as Day 0) and on the last day of the treatment period (i.e., Day 10 of the treatment period) using PAI (Figure 4A), comparison of the baseline percent oxygen saturation (%sO_2_) values between A549 and A549shHTATIP2 tumors showed that the mean baseline %sO_2_ value of A549shHTATIP2 tumors was 17% lower than that of A549 tumors (*p* < 0.01, *n* = 12) (Figure 4B). Comparison of the %sO_2_ values between A549 and A549shHTATIP2 tumors after the 10-day vehicle or sorafenib treatment showed that the mean %sO_2_ value of vehicle-treated A549shHTATIP2 tumors was still significantly lower (by 13%) than that of vehicle treated A549 tumors (*n* = 6, *p* < 0.01), whereas no significant difference in the mean %sO_2_ values was found between sorafenib-treated A549 and A549shHTATIP2 tumors (*n* = 6, *p* > 0.05) (Figure 4B). Comparison of the mean %sO_2_ values before and after sorafenib treatment indicated that sorafenib treatment significantly reduced the mean %sO_2_ value in A549 tumors by 20% (*p* < 0.05; paired samples *t*-test), but had no significant effect on the mean %sO_2_ value in A549shHTATIP2 tumors (*p* > 0.05) (Figure 4B). To examine if the rapid tumor growth and reduced intratumor oxygenation were associated with the decreased sorafenib distribution in A549shHTATIP2 tumors, it was determined sorafenib concentrations in plasma and tumor samples collected at 4 h after dosing on the last day of the experiment, but found no significant difference in the tumor-to-plasma sorafenib concentration ratio between A549 and A549shHTATIP2 tumors (1.05 ± 0.41 versus 0.74 ± 0.26; *p* > 0.05; *n* = 6). Taken together, the data demonstrated that the absent expression of HTATIP2 in A549 cells resulted in an aggressive tumor phenotype, marked by rapid growth, poor oxygenation and decreased sensitivity to sorafenib treatment, which was not attributable to the decreased sorafenib distribution in tumor.

### 2.4. Reduced HTATIP2 Expression Promoted Cell Proliferation In Vivo Despite of Stalled Tumor Angiogenesis

To examine the aggressive phenotype of A549shHTATIP2 tumors at the cellular level and identify the molecular pathway involved, tumor samples collected from the *in vivo* study were subjected to immunofluorescent staining, Western blot analysis and real-time PCR analysis (Figure 5 and Figure 6). Immunofluorescent staining of A549 and A549shHTATIP2 tumor cryosections was carried out for the cell proliferation marker Ki67, the endothelial cell marker CD31 and the hypoxia marker pimonidazole (Figure 5A,B). In the vehicle control group, the mean Ki-67 proliferation index value of A549shHTATIP2 tumors was significantly higher than that of the A549 tumors (24.8% versus 12.9%; *p* < 0.01), whereas the mean MVD in A549shHTATIP2 tumors was significantly lower than that in the A549 tumors (1.5% versus 3.9%, *p* < 0.01) (Figure 5B). In the sorafenib treatment group, the mean Ki-67 proliferation index value of A549shHTATIP2 tumors was significantly higher than that in the A549 tumors (11.4% versus 6.2%; *p* < 0.01), while no significant difference in MVD was found between A549shHTATIP2 and A549 tumors (1% versus 1.3%, *p* > 0.05) (Figure 5B). Comparison between vehicle- and sorafenib-treated tumors showed that the mean Ki-67 proliferation index values in both sorafenib-treated A549 and A549shHTATIP2 tumors were significantly lower than those in their respective vehicle controls (*p* < 0.01 for both), indicating that the rapidly proliferating cells in the A549shHTATIP2 tumors were sensitive to the short-term sorafenib treatment. The mean MVD in sorafenib-treated A549 tumors was significantly decreased as compared with that in the vehicle control A549 tumors (*p* < 0.01). The MVD in sorafenib-treated A549shHTATIP2 tumors was also significantly decreased (*p* < 0.05) but to a lesser extent as compared with that in the sorafenib-treated A549 tumors (68% versus 29% of control values). Additionally, the existence of tumor hypoxia in all tumor xenograft samples was indicated by the positive pimonidazole staining. This observation was in line with the reported extensive hypoxia in A549 subcutaneous tumor model [36]. Quantitative evaluation of pimonidazole stained tumor sections was not feasible due to the considerable difference in pimonidazole staining pattern among tumors. Overall, the Ki67 and CD31 immunofluorescent staining data obtained from the vehicle control groups were consistent with the relatively rapid growth and low oxygenation observed in the A549shHTATIP2 tumors. Although the inhibitory effect of short-term sorafenib treatment on cell proliferation in the A549shHTATIP2 tumors was comparable to that in the A549 tumors, the antiangiogenic effect of sorafenib was absent in the A549shHTATIP2 tumors. This observation raises a question: how is the growth of A549shHTATIP2 tumors sustained with limited blood supply?

### 2.5. Absent Expression of HTATIP2 Resulted in Upregulated HIF2α Expression, Enhanced β-Catenin/c-Myc/MCL-1 Signaling and Elicited EMT in Response to Sorafenib Treatment

To explore the molecular mechanism involved in the modulatory effect of the absence of HTATIP2 expression on tumor adaptation to hypoxic microenvironment with and without sorafenib treatment, the expression levels of important mediators and effectors in hypoxia-elicited pathways in the A549 and A549shHTATIP2 tumor samples using immunoblot and real-time PCR analyses were examined (Figure 6 and Appendix A). Results of immunoblot analysis showed that, in the vehicle group, the protein levels of HIF2α, MCL-1 and β-catenin in the A549shHTATIP2 tumors were significantly increased as compared with that in the A549 tumors (*p* < 0.01 for HIF2α and *p* < 0.05 for both MCL-1 and β-catenin) (Figure 6A,B). Sorafenib treatment significantly decreased the protein levels of c-Myc (*p* < 0.01 for both) and β-catenin (*p* < 0.05 for A549 and *p* < 0.01 for A549shHTATIP2 tumors) in A549 and A549shHTATIP2 tumors. It also appeared to decrease the protein levels of MCL-1and phospho-Akt in both tumor types although only the reduction of those proteins in A549shHTATIP2 tumors reached statistical significance (*p* < 0.05 for MCL-1, and *p* < 0.01 for phospho-Akt), suggesting that the response of A549 and A549shHTATIP2 tumors to sorafenib treatment involves the same set of signaling pathways that are affected to a different extent (Figure 6A,B and Appendix A). These data were in line with the Ki67 immunostaining results (Figure 5B), indicating that the response of A549shHTATIP2 xenografts to short-term sorafenib treatment was comparable to that of their parental counterparts. To evaluate the potential influence of the absent expression of HTATIP2 on the EMT in tumor cell-derived xenografts, the E-cadherin and vimentin protein levels were analyzed. Although no statistically significant difference in the E-cadherin and vimentin expression was found between vehicle control A549 and A549shHTATIP2 tumors, the expression of E-cadherin in sorafenib-treated A549shHTATIP2 tumors was significantly decreased by 24% compared to the vehicle-treated A549shHTATIP2 tumors (*p* < 0.01), implicating that sorafenib treatment significantly disrupts intercellular contacts in the A549shHTATIP2 tumors. The expression of vimentin in sorafenib-treated A549shHTATIP2 tumors was significantly increased by 21% compared to their sorafenib-treated parental counterparts (*p* < 0.05) (Figure 6A,B), suggesting that the absence of HTATIP2 expression increases the susceptibility of A549 tumors to sorafenib-activated EMT process. Consistent with the immunoblot data, results of the real-time PCR analysis demonstrated that the mean *HIF2α* mRNA level in the vehicle control A549shHTATIP2 tumors was significantly increased as compared with that in the vehicle control A549 tumors, while sorafenib treatment significantly decreased the *c-Myc* mRNA levels in A549 and A549shHTATIP2 tumors (Figure 6C). Moreover, in the vehicle group, the mean *VEGFA* mRNA level in the A549shHTATIP2 tumors was significantly lower than in that their parental counterparts (*p* < 0.05), which was in agreement with the MVD data (Figure 5B and Figure 6C), whereas the mean *GLUT-1* mRNA level in A549HTATIP2 tumors was higher than that in the A549 tumors (*p* < 0.05) (Figure 6C). Furthermore, treatment with sorafenib for 10 days resulted in a significant decrease in the *HIF1α* mRNA level in both A549 and A549shHTATIP2 tumors (*p* < 0.01 for both), a significant decrease in the *HIF2α* mRNA level (*p* < 0.01) and a significant increase in the *VEGFA* mRNA level (*p* < 0.05) in the A549shHTATIP2 tumors (Figure 6C). Overall, the data suggest that the absence of HTATIP2 expression fine-tunes tumor adaptation to hypoxia and subsequent response to treatment through HIF2α and its target genes.

### 2.6. Metabolomic Profiling of A549 and A549shHTATIP2 Xenografts

To further explore the mechanism that accounts for the rapid growth with stalled angiogenesis observed in the A549shHTATIP2 tumors, we employed the metabolomics approach to identify changes in the pattern of tumor metabolism that is directly connected with the hypoxia signaling system [37]. The quantities of 248 metabolites were determined in 4 vehicle control A549 and 6 vehicle control A549shHTATIP2 tumor homogenates using the LC-MS/MS (Appendix A). Metabolites which levels in A549shHTATIP2 tumors were increased by more than 100% compared with those in the control A549 tumors were homocysteine, cysteine, 5-aminoimidazole-4-carboxamide-1-β-riboside (AICA-riboside), and long-chain acylcarnitines including decanoyl-L-carnitine, octanoyl-L-carnitine, lauroyl-L-carnitine, myristoyl-L-carnitine, and stearoyl-L-carnitine (Table 1). Homocysteine and cysteine are central to cysteine and methionine metabolism [38,39], while carnitine derivatives are crucial mediators of metabolic plasticity in cancer cells [40]. Among those metabolites, the mean levels of homocysteine (*p* < 0.01), myristoyl-L-carnitine (*p* = 0.01) and octanoyl-L-carnitine (*p* < 0.05) were significantly increased in A549shHTATIP2 tumors as compared with those in A549 tumors (Table 1). Metabolites whose levels were significantly decreased or decreased by more than 50% in A549shHTATIP2 tumors included those that are essential to the tricarboxylic acid (TCA) cycle (e.g., ATP, nicotinamide adenine dinucleotide phosphate (NADP), ADP, coenzyme A (CoA), acetyl-CoA and succinyl-CoA), to the DNA and RNA syntheses (e.g., adenosine diphosphate ribose (ADP ribose), β-nicotinamide D-ribonucleotide and guanosine), to the fatty acid catabolism (e.g., malonyl-CoA, propionyl-CoA and ethanolamine) and to the glutaminolysis (e.g., glutamine), as well as glutathione, N-acetylornithine and thiamine. In particular, the mean levels of NADP, CoA, acetyl-CoA and succinyl-CoA in A549shHTATIP2 tumors were significantly decreased by 61% (*p* < 0.05), 75% (*p* < 0.01), 57% (*p* < 0.01) and 50% (*p* < 0.01), respectively, and those of propionyl-CoA and ethanolamine by 60% and 51%, respectively (*p* < 0.01 for both), and that of glutamine by 48% (*p* < 0.01), and those of β-nicotinamide D-ribonucleotide and guanosine by 54% (*p* < 0.05) and 53% (*p* < 0.01), respectively, and those of glutathione and thiamine by 68% and 59%, respectively (*p* < 0.01 for both) (Table 1). Taken together, the data suggest that the absence of HTATIP2 expression results in tumor metabolic rewiring that aggravates the aggressive tumor growth.

## 3. Discussion

Although accumulating evidence has demonstrated that HTATIP2 acts as a tumor suppressor, of which reduced or absent expression is associated with increased susceptibility to tumorigenesis [41] and enhanced tumor invasion and metastasis [20], the role of HTATIP2 in hypoxia-regulated adaptive responses that promote tumor progression is unclear. Results from the *in vitro* study provided limited information on the modulatory effect of HTATIP2 knockdown on tumor cell response to hypoxia due to the inability of *in vitro* systems to recapitulate the complex cellular interactions in the tumor microenvironment that determine the phenotypic behavior of tumor cells [42]. Since subcutaneous lung tumor xenografts have been shown to exhibit significant hypoxia when compared with orthotopic lung tumor xenografts and spontaneous lung tumors [36], a subcutaneous A549 human lung adenocarcinoma xenograft model was used in the present study to explore the modulatory effect of the absent expression of HTATIP2 on tumor adaptation to hypoxia and associated mechanisms. The presence of tumor hypoxia was confirmed by the positive pimonidazole staining of the tumor sections (Figure 5A) and reduced intratumoral blood oxygen saturation levels observed using PAI (Figure 4).

With the double subcutaneous xenograft model, we observed a significantly rapid growth of A549shHTATIP2 tumors. Further analyses revealed a stalled tumor angiogenesis, marked by decreased tumor oxygenation (Figure 4), CD31 immunostaining (Figure 5A,B) and *VEGFA* mRNA expression (Figure 6C). Moreover, we observed a significant increase in HIF2α protein and mRNA levels in the vehicle-treated A549shHTATIP2 xenografts as compared with those in their counterpart parental xenografts. Our observation of the decreased tumor vasculature in A549shHTATIP2 tumors with upregulated HIF2α expression was discordant with an early in vitro study showing that the proliferation and migration of human umbilical vein endothelium cells (HUVECs), human lung microvascular endothelial cells (HLMVECs) and vascular smooth muscle cells (VSMCs) were inhibited when those cells were treated with the culture media conditioned by HTATIP2-expressing small cell lung carcinoma (SCLC) cells [43]. This discrepancy may be attributed to the different tumor cell lines (SCLC versus lung adenocarcinoma cells) and model systems (*in vitro* versus *in vivo*) used. Both HIF1α and HIF2α are known to play an important role in tumor angiogenesis. HIF1α is considered the fundamental initiator of tumor angiogenesis, which initiates the process through regulating the key angiogenic factor vascular endothelial growth factor A (VEGFA) via a hypoxia-response element (HRE) present in the *VEGFA* promotor [44]. HIF2α, which is abundantly expressed in endothelial cells, promotes endothelial adherens junction integrity and enhances endothelial barrier [45]. In this study, we demonstrated that treatment with sorafenib for 10 consecutive days significantly decreased the tumor microvessel density in A549 parental tumors but not in A549shHTATIP2 tumors, while there was no significant difference in sorafenib concentrations between A549 and A549shHTATIP2 tumors. Since antiangiogenic agents such as sorafenib can transiently prune the immature newly formed microvessels while sparing the relatively efficient vessels [46], our data implicate that upregulation of HIF2α expression and downregulation of *VEGFA* expression in A549shHTATIP2 tumors result in reduced tumor neovascularization but improved functional vascular properties. In addition, it has been demonstrated that treatment with VEGF in human endothelial cell lines resulted in increased *HTATIP2* and *HIF1α* gene expression and decreased *HIF2α* gene expression [47]. However, it remains ambiguous whether the observed increase in *VEGFA* mRNA levels in A549shHTATIP2 cells is associated with the decreased HIF2α expression upon treatment with sorafenib or due to blockade of tumor vasculature by angiogenic inhibitors that leads to the compensatory release of angiogenic inducers, such as VEGFA [48].

It has been reported that knockdown or overexpression of HIF2α negatively affected HTATIP2 protein expression in HCC cells under CoCl_2_ (400 μM) treatment, but not the other way around [31]. Further chromatin immunoprecipitation (ChIP) analysis has revealed that HIF2α binds to the HTATIP2 promotor, suggesting that the inverse relationship between HTATIP2 and HIF2α is a result of HTATIP2 being negatively regulated by HIF2α [31,32]. In this study, although no significant difference in the HIF2α protein level was found between A549shNT and A549shHTATIP2 cells cultured under hypoxic (0.5% O_2_) condition, there was a significant increase in the expression levels of both HIF2α mRNA and protein in the vehicle-treated A549shHTATIP2 xenograft tumors as compared with their parental counterparts (*p* < 0.01 for both) (Figure 6). Our co-immunoprecipitation results not only confirmed the interaction between HTATIP2 and HIF2α, but also demonstrated the presence of interaction between HTATIP2 and c-Myc and the absence of interaction between HITATIP2 and HIF1α (Figure 2C). HIF1α and HIF2α have been shown to have opposite effects on renal cell carcinoma (RCC) xenograft tumor growth [49], which were associated with their antagonistic effects towards the c-Myc oncogenic activity [50]. HIF2α promotes RCC tumor growth by increasing c-Myc transcription activity that promotes cell cycle progression and cellular proliferation, whereas HIF1α antagonizes c-Myc function by displacing c-Myc from p21^cip1^ promotor, leading to the activation of the cyclin-dependent kinase (CDK) inhibitor p21^cip1^ that causes cell cycle arrest in G1 phase or in the G2/M transition after DNA damage [51,52]. In addition, HIF1α can repress c-Myc activity by activating transcription of the gene encoding MXI1, a c-Myc antagonist, and promoting MXI1-independent and proteasome-dependent degradation of c-Myc [53]. In this study, there was no significant difference in the mRNA and protein expression of both HIF1α and MXI1 between A549 parental and A549shHTATIP2 tumors, suggesting the absence of HTATIP2 expression is unlikely to have any effect on the activity of either HIF1α or MXI1 (Figure 6A,B).

Although our data demonstrated the significant increase in HIF2α mRNA and protein levels in A549shHTATIP2 xenograft tumors, the increase in c-Myc mRNA and protein expression was not statistically significant (*p* > 0.05 for both). Nonetheless, we observed a significant increase in MCL-1 protein expression (*p* < 0.05) in the vehicle-treated A549shHTATIP2 tumors as compared with their parental counterparts (Figure 6A,B). MCL-1 is an anti-apoptotic BCL-2 family protein that inhibits apoptosis by preventing cytochrome c release [54,55]. The expression of MCL-1 can be regulated by c-Myc, which binds to the E-box in the promoter of the *MCL-1* gene and thus controls MCL-1 transcription [56]. Clinical studies have demonstrated that MCL-1 is often overexpressed in non-small cell lung cancer (NSCLC) [57,58], and MCL-1 overexpression together with c-Myc overexpression is correlated with poorer overall survival of patients with NSCLC [59]. Our data showed that the significant decrease in c-Myc protein and mRNA levels upon sorafenib treatment in A549 parental and A549shHTATIP2 tumors (*p* < 0.01 for all) was accompanied by a decrease in MCL-1 protein levels although the statistical significance was reached only in A549shHTATIP2 tumors for MCL-1 (*p* < 0.01) (Figure 6A,B). This observation was in line with a previous *in vitro* study demonstrating that the c-Myc-knockdown gastric cancer cells exhibited decreased MCL-1 protein and mRNA expression levels [56]. Since c-Myc is known to concomitantly induce cell proliferation and apoptosis, the significantly increased MCL-1 levels observed in HTATIP2-deficient A549 tumors is likely to contribute to the rapid tumor growth by increasing the threshold of c-Myc activity required for initiation of apoptosis which in turn facilitates c-Myc-stimulated cell proliferation [59,60].

In addition to MCL-1, the β-catenin protein level in the vehicle-treated A549shHTATIP2 tumors was also significantly higher than that in their parental counterparts (*p* < 0.05) (Figure 6A,B). β-Catenin is a canonical effector of Wnt signaling responsible for cell differentiation, proliferation and survival [61]. Stabilized β-catenin forms a complex with T cell factor/lymphoid enhancer factor (TCF/LEF) in the nucleus, leading to the activation of proliferative Wnt target genes, including c-Myc [62], and induction of EMT activator zinc finger E-box-binding homeobox 1 (ZEB1) [63]. Moreover, the Wnt/β-catenin pathway is linked to MCL-1 through c-Myc [64,65]. Both HIF1α and HIF2α interact with β-catenin, but at different sites. HIF1α inhibits β-catenin activity by competing with TCF-4 for direct binding to β-catenin [66], whereas HIF2α enhances the transcriptional activity of β-catenin by recruiting P-300 through its transactivation domain [67]. Our data suggest that the increased HIF2α expression is associated with the upregulation of β-catenin in A549shHTATIP2 tumors, which in turn contributes to the relatively rapid tumor growth and potential EMT. Upon the treatment with sorafenib, the β-catenin protein expression level decreased significantly in both A549 parental and A549shHTATIP2 tumors, coinciding with the significant decrease in c-Myc mRNA and protein levels (Figure 6). Since β-catenin is associated with c-Myc by functioning as a transcription factor to activate c-Myc [62], this result suggests that short-term sorafenib treatment inhibits the c-Myc-stimulated cell proliferation through decreasing the activity of β-catenin in both A549 and A549shHTATIP2 tumors. However, unlike the A549 tumors, sorafenib treatment resulted in not only a significantly decreased β-catenin protein level but also a significantly decreased E-cadherin protein level in A549shHTATIP2 tumors (*p* < 0.01 for both) as compared with the vehicle-treated A549shHTATIP2 tumors. It is speculated that the decreased β-catenin levels in A549shHTATIP tumor tissue homogenates is in part attributable to the decreased membranous and cytoplasmic β-catenin levels associated with the liberation of β-catenin from the cytoplasmic tail of E-cadherin [68] followed by the nuclear translocation of β-catenin that potentially facilitates the transcriptional changes associated with tumor metastasis [69,70]. In addition, the observed significant increase in vimentin protein expression in sorafenib-treated A549shHTATIP2 tumors compared to the sorafenib-treated A549 tumors (*p* < 0.05) (Figure 6A,B) reinforced our belief that the absence of HTATIP2 expression increases the susceptibility of A549 tumors to sorafenib-activated EMT process. In the future, it will be interesting to explore HIF2α-regulated metastasis signaling cascade associated with the adaptation of A549shHTATIP2 tumors to long-term sorafenib treatment.

One of the central characteristics of tumor malignancy is metabolic reprogramming in response to hypoxic stress to meet the demands of proliferation [71]. To examine further whether the HTATIP2-deficient A549 tumors exhibit any peculiar metabolic alterations that support their rapid growth without increased vascularity, we determined the difference in metabolic profiles between vehicle-treated A549 parental and A549shHTATIP2 tumors using the LC-MS/MS-based metabolomic approach. We detected more than 1-fold increase in the levels of homocysteine, cysteine and several carnitine derivatives, and a net decrease in the ATP production from TCA cycle due to the reduced glutaminolysis and fatty acid catabolism despite a possible increase in glycolysis manifested by the increased *GLUT1* mRNA levels in A549shHTATIP2 tumors (Table 1 and Figure 6C). Our observation suggests that the HTATIP2-deficient A549 tumors are not dependent on glutamine and fatty acid as a carbon source to replenish TCA cycle intermediates, but rather exploit alternative *metabolic* pathways for survival and proliferation (the so-called metabolic plasticity). This postulation is supported by several lines of evidence indicating that cancer cells can obtain exogenous non-essential amino acids, such as cysteine and serine, from their surroundings to fuel their anabolic metabolism [39,72], and re-methylation of homocysteine is part of the methionine cycle associated with the folate cycle in serine metabolism that integrates nutrient status and availability [38]. Moreover, the carnitine system is involved in the cross-talk with lipid metabolism and aerobic glycolysis that finely triggers the metabolic plasticity of cancer cells [40]. Although it has been reported that genes involved in cysteine and methionine metabolism were significantly upregulated in primary human microvascular endothelial cells (hMVECs) exposed to prolonged hypoxia in the presence of VEGFA/TNFα [73], the mechanism by which HIF1α or HIF2α regulates the cysteine metabolism and carnitine system remains to be determined.

## 4. Materials and Methods

### 4.1. Reagents

Sorafenib base and sorafenib tosylate were purchased from LC Laboratories (Woburn, MA, USA). For the *in vitro* study, sorafenib base was dissolved in dimethyl sulfoxide (DMSO). To prepare the stock solutions for the *in vivo* study, sorafenib tosylate was dissolved in Cremophor EL/ethanol (50:50, *v*/*v*) at 4× concentration and then diluted with sterile water to 1× concentration. MISSION^®^ shRNA lentiviral transduction particles containing the shRNA sequence targeting the coding region of HTATIP2 gene (TRCN0000280445) and MISSION^®^ TRC2 pLKO.5-puro non-target shRNA control transduction particles (Catalog number: SHC216V) were purchased from Millipore Sigma (St. Louis, MO, USA). All other chemicals, solvents and reagents were obtained from commercial sources.

### 4.2. Animals

Male athymic nude mice (Hsd: Athymic Nude-Foxn1nu; 5–6 weeks old) were purchased from Envigo (Indianapolis, IN, USA). All animal experiments were approved by the Institutional Animal Care and Use Committee (IACUC) (Project ID: IS00005995) and performed according to the National Institute of Health (NIH) guidelines.

### 4.3. Cell Lines, Cell Culture and Stable Short Hairpin RNA (shRNA) Knockdown

Three human lung adenocarcinoma cell lines, A549, NCI-H358 and NCI-H1915, and one human squamous cell carcinoma cell line, SK-MES-1, were purchased from the American Type Culture Collection (ATCC). The A549 cell line was cultured in a mixture of Dulbecco’s Modified Eagle’s Medium (DMEM)/Ham’s F12 at a ratio of 1:1 (Corning^TM^ 10-092-CV. Thermo Fisher Scientific, Waltham, MA, USA) supplemented with 10% heat-inactivated fetal bovine serum (FBS) (Gibco^TM^#10082147. Thermo Fisher Scientific, Waltham, MA, USA). NCI-H358, NCI-H1915 and SK-MES-1 cell line were cultured in DMEM medium (Corning^TM^ 10-013-CV. Thermo Fisher Scientific, Waltham, MA, USA) supplemented with 10% heat-inactivated FBS. Antibiotics were added to the cell culture media (100 units/mL penicillin and 100 µg/mL streptomycin) to prevent contamination. All cells were maintained at 37 °C in an atmosphere of humidified air with 5% CO_2_. Since HTATIP2 protein expression was not detectable in NCI-H358 and NCI-H1915 cell lines and SK-MES-1 cells rapidly lost proliferative potential and entered a senescent state after the lentivirus-mediated transduction of shRNA targeting HTATIP2, only A549 cell line was found suitable for the study (Appendix A). The stable knockdown of shRNA-targeting genes was carried out by transducing A549 cells with lentiviral transduction particles containing HTATIP2-targeting shRNA (A549shHTATIP2) or a non-target scrambled shRNA (A549shNT) in the presence of hexadimethrine bromide (8 μg/mL) followed by selection with 5 μg/mL of puromycin for 2 weeks. The selected puromycin-resistant cells were maintained under the same culture condition as that described for the A549 parental cells. Reduction of HTATIP2 protein expression was confirmed by Western blot analysis.

### 4.4. Hypoxia

Hypoxia *in vitro* (0.5% O_2_, 5% CO_2_ and 94.5% N_2_) was achieved and maintained using a hypoxia incubator chamber (Catalog number# 27310. STEMCELL Technologies Inc., Cambridge, MA, USA) receiving gas from a custom-mixed gas tank (Airgas Inc., Radnor, PA, USA).

### 4.5. Wound-Healing Assay

The migratory potential of A549shNT and A549shHTATIP2 cells that were maintained in serum-free DMEM/F12 medium under hypoxic and normoxic conditions for up to 48 h was assessed using the wound-healing assay as described previously with little modification [74,75]. The wound areas on cell monolayers were created using a sterile 10 μL (P10) pipette tip.

### 4.6. Cell Invasion Assay

To assess the invasive ability of A549shNT and A549shHTATIP2 cells, a Matrigel invasion assay was carried out under hypoxic and normoxic conditions in 24-well modified Boyden chambers with 8-μm pore polycarbonate filters (Corning Costar, Cambridge, MA, USA) that were pre-coated with (invasion) or without (migration) 300 μg/mL of Growth Factor Reduced Corning^®^ Matrigel^®^ Matrix (Cat. No. 354230. Corning Inc. Tewksbury, MA, USA) as described previously with little modification [75].

### 4.7. Co-Immunoprecipitation

Co-immunoprecipitation assay was used to identify the interaction between HTATIP2 and HIF1α or HIF2α. Briefly, an antibody specific for HTATIP2, HIF1α or HIF2α was used to immunoprecipitate the target protein from the whole cell lysate of A549 cells cultured under the hypoxic condition for 24 h. The immune-complexes were recovered using Protein A agarose beads (Catalog #9863. Cell Signaling Technology, Danvers, MA, USA) according to the manufacturer’s protocol, and then resolved by sodium dodecyl sulfate – polyacrylamide gel electrophoresis (SDS–PAGE). The amounts of immunoprecipitated protein and co-immunoprecipitated associated proteins were detected by Western blot analysis using anti-HIF1α, anti-HIF2α, anti-c-Myc and anti-HTATIP2 antibodies (Cell Signaling Technology, Danvers, MA, USA).

### 4.8. In Vitro Cytotoxicity Assay

The MTT (3-[4,5-dimethylthiazole-2-yl]-2,5-diphenyl-tetrazolium bromide) assay was performed to evaluate the impact of HTATIP2 knockdown on A549 cell response to sorafenib treatment under hypoxic and normoxic conditions. Briefly, A549shNT and A549shHTATIP2 cells were seeded in 96-well plates at a density of 3 × 10^3^ cells/well and allowed to attach overnight. On the next day, culture media containing either vehicle control (0.5% DMSO) and sorafenib (13 nM–50,000 nM) were added to appropriate wells. After the cells were treated for 72 h under hypoxic or normoxic conditions, 5 μL of 5 mg/mL of MTT in PBS was added to each well and individual plates were incubated for 2 h at 37 °C followed by the addition of 100 μL of DMSO to each well and incubation at room temperature in the dark for 2 more hours. Optical densities were measured at 570 nm with a SpectraMax 190 microplate reader equipped with SoftMax Pro version 4.3.1 software (Molecular Devices, Sunnyvale, CA, USA). The viability of treated cells was expressed as a percentage of vehicle control cultures. Concentrations of sorafenib required for 50% inhibition of cell growth (i.e., IC50) as compared with the control cells were calculated by nonlinear fitting of the experimental data obtained from multiple independent experiments performed in duplicates or triplicates using the GraphPad Prism version 5.0 program (GraphPad Software, Inc. La Jolla, CA, USA).

### 4.9. In Vivo Study Protocol

Since the subcutaneous human A549 lung carcinoma xenografts have been shown to demonstrate significant hypoxia as compared with the orthotopic A549 xenografts and the spontaneous murine lung tumors [36], the impact of absent HTATIP2 expression on tumor cell adaptation to hypoxic microenvironment that promotes changes in favor of tumor growth was examined in a double subcutaneous xenograft model [75]. Briefly, each male athymic nude mouse was inoculated subcutaneously with 5 × 10^6^ A549 parental cells in the left dorsal flank and the same number of A549shHTATIP2 cells in the opposite flank. Tumor growth was monitored once a week with the volume calculated as 0.5 × length × width^2^. Six weeks after tumor inoculation, tumor-bearing animals were randomly divided into control and sorafenib groups (*n* = 6 for each). Each animal received once-daily oral administration of vehicle control or 40 mg/kg sorafenib for 10 consecutive days starting on Day 43. Tumor-bearing animals were subjected to photoacoustic imaging (PAI) that enables the measurement of intratumoral oxygen saturation (sO_2_) on Day 42 (Day 0 of the treatment period) and at approximately 2 h after the last dose of the vehicle or sorafenib on Day 52 (Day 10 of the treatment). Immediately after the PAI on Day 52, each animal was given a tail-vein injection of 60 mg/kg pimonidazole (Natural Pharmacia International Inc., Burlington, MA, USA), which allows the assessment of intratumoral hypoxic regions using immunofluorescence staining. Animals were euthanized 1.5 h after the administration of pimonidazole. Plasma was separated from whole blood by centrifugation and then stored at −80 °C before subjected to drug analysis using the established high-performance liquid chromatography (HPLC) method [75]. Tumors were excised and weighed. Each tumor was divided into several pieces for snap-freezing on dry ice or preserving in Trizol^TM^ reagent (Invitrogen #15596026. Thermo Fisher Scientific, Waltham, MA, USA), and then stored at −80 °C before further analyses.

### 4.10. Measurement of Intratumoral Oxygen Saturation Using Real-Time In Vivo Photoacoustic Imaging

Each tumor-bearing mouse was anesthetized with isoflurane (1.5%, delivered in oxygen) and placed prone on a heated physiological monitoring stage. Clear ultrasound (US) gel (Parker Laboratories. Fairfield, NJ, USA) was used to facilitate the transmission of sound waves from the transducer to the animal. The intratumoral sO_2_ was measured using the Vevo LAZR-X real-time *in vivo* PAI system Vevo LAB version 3.1.1 (FUJIFILM VisualSonics, Inc. Toronto, ON, Canada). The tumors were scanned in 3 dimensions (3D) via a 256-element linear array LZ250 transducer with the axial resolution of 75 µm and broadband frequency of 13 to 24 MHz and a center operating frequency of 21 MHz. Co-registered 3D photoacoustic and ultrasound images were acquired with 10 nm incremental steps over the entire tumor mass. Oxygen saturation was calculated as the percentage of oxygenated hemoglobin relative to total hemoglobin based on dual-wavelength photoacoustic imaging at 750 and 850 nm wavelength.

### 4.11. Immunofluorescence Staining

Frozen subcutaneous tumor samples collected from the *in vivo* study were cryosectioned at 10 μm and fixed in 4% *paraformaldehyde (PFA) for* 15 min and then subjected to immunofluorescence staining as described previously [76]. In brief, tumor sections were incubated overnight at 4 °C with a rat anti-mouse CD31 antibody (1:200; BD Pharmingen Catalog number: 550274; BD Biosciences, San Jose, CA, USA), a rabbit anti-Ki67 antibody (1:100; Catalog number: ab16667; Abcam, Cambridge, MA, USA) and a rabbit anti-pimonidazole antibody (1:100; Catalog number: PAb2627AP; Hypoxyprobe, Inc., Burlington, MA, USA) followed by 1-h incubation at room temperature in dark with Alexa Fluor 488 conjugated goat anti-rat IgG (1:200) or Alexa Fluor 568 conjugated goat anti-rabbit IgG (1:200) (Invitrogen). The microvessel density (MVD) was determined by CD31 immunofluorescence staining and expressed as a percentage of CD31-positive area relative to the area of optical field [77]. The Ki67 proliferation index was expressed as a percentage of Ki67 positive cells per total number of counted cells [77]. Areas with extensive necrosis were avoided. Images were processed with the ImageJ version 1.52a software from the NIH (Bethesda, MD, USA) and available at https://imagej.nih.gov/ij/.

### 4.12. Semi-Quantitative Western Blot Analysis

Cell lysate samples were prepared from cultured A549shNT and A549shHTATIP2 cells that were treated with vehicle (0.5% DMSO) or sorafenib (10 μM) and maintained in DMEM medium with 4.5 g/L glucose and L-glutamine and without sodium pyruvate (Gibco # 11965092. Thermo Fisher Scientific, Waltham, MA, USA) and supplemented with 0.5% heat-inactivated FBS under hypoxic or normoxic conditions for 24 h. Tumor tissue homogenate samples were prepared from tumor samples collected from the *in vivo* studies. Cell lysates and tumor tissue homogenates were reduced and denatured by boiling the samples in sample buffer containing dithiothreitol (DTT) and sodium dodecyl sulfate (SDS) at 100 °C for 5 min before loaded onto SDS-polyacrylamide gel electrophoresis (PAGE) gels as described previously [77]. The amount of total protein loaded per lane was about 20 and 75 μg for cell lysates and tumor tissue homogenates, respectively. The primary antibodies used in the Western blot analysis are listed in Appendix A. Western blot densitometric analysis was carried out using the ImageJ software (https://imagej.nih.gov/ij/). Normalization for loading differences was achieved by dividing the densitometric values for individual bands by the densitometric values for β-actin in the same lane. Densitometry values of individual blots were calculated as:(1)$$100\times {\left[\mathrm{Density}\right]}_{\mathrm{target}\;\mathrm{protein}\;\mathrm{in}\;\mathrm{sample}}/{\left[\mathrm{Density}\right]}_{\beta -\mathrm{Actin}}/{\left[\mathrm{Density}\right]}_{\mathrm{target}\;\mathrm{protein}\;\mathrm{in}\;\mathrm{positive}\;\mathrm{control}}$$

### 4.13. Quantitative Real-Time Polymerase Chain Reaction (PCR)

One microgram of total RNA extracted from tumor tissues with Trizol^TM^ reagent (Invitrogen #15596026) was reverse transcribed into complimentary DNA (cDNA) using random hexamer primers and AMV reverse transcription reagents as per the manufacturer’s protocol (Promega, Madison, WI, USA). Then, 25 ng of cDNA template was subjected to the quantitative real-time PCR analysis for human *HTATIP2*, *HIF1α*, *HIF2α*, *c-Myc*, *MXI1*, *VEGFA*, *PDGFB*, *GLUT1*, and *GAPDH* (the endogenous control) genes. Sequences of primers used for the real-time PCR are shown in Appendix A. The PCR analyses were performed using the CFX96 Touch Real-Time PCR detection system (Bio-Rad Laboratories, Hercules, CA, USA). The reactions were carried out in duplicate using the iTaq™ Universal SYBR^®^ Green Supermix (Catalog number: 1725121; Bio-Rad Laboratories, Inc. Hercules, CA, USA) according the manufacturer’s protocol. The normalized expression was calculated using the 2^−ΔΔCt^ method as previously described [78].

### 4.14. Determination of Sorafenib Concentrations in Plasma and Tumor Tissue Homogenates using High Performance Liquid Chromatography

A high performance liquid chromatography (HPLC) method was developed and validated for the determination of sorafenib concentrations in mouse plasma and xenograft tumor tissues. Briefly, plasma and tumor tissue homogenate (tissue:MilliQ water = 1:5, *w*/*v*) samples were deproteinated by adding three volumes of methanol containing 5 μg/mL of phenprocoumon (the internal standard, IS) followed by the centrifugation at 14,000 rpm for 10 min. Ten μL aliquots of the supernatants were injected onto the reversed-phase HPLC system with a diode array detector. The chromatographic separation was achieved on an octadecylsilane bonded silica column (Luna^®^ 3 μm C18, 50 × 4.6 mm, Phenomenex) at room temperature. Isocratic elution was employed using 55% acetonitrile containing 10 mM ammonium acetate and 0.1% formic acid. The IS and sorafenib were detected at 325 and 280 nm, respectively. Retention time was about 1.5 and 2.5 min for IS and sorafenib, respectively. Standard curves of sorafenib were linear within the ranges of 41–5000 and 123–10,000 ng/mL in plasma and tumor homogenates, respectively (*r*^2^ > 0.99 for both).

### 4.15. Liquid Chromatography Tandem Mass Spectrometry (LC-MS/MS) Based Targeted Metabolomics

Metabolites in vehicle-treated A549 and A549shHTATIP2 xenograft tumors were quantitatively profiled using an LC-MS/MS-based targeted metabolomics platform as described previously [79]. In brief, metabolites in xenograft tumor homogenates were extracted with methanol. The metabolite extracts were dried in a CentriVap vacuum evaporator (Labconco, Kansas City, MO, USA). Following centrifugation of the reconstituted dried extracts, the supernatant was subjected to quantitative metabolomic profiling performed on an AB SCIEX QTRAP 6500 LC-MS/MS system (SCIEX, Framingham, MA, USA). Metabolomics data analyses were carried out using the MetaboAnalyst web-based statistical package (http://www.metaboanalyst.ca/). To meet the normality assumption, individual metabolites concentrations were log-transformed and then autoscaled (mean-centered and divided by the standard deviation of each metabolite) [79,80].

### 4.16. Statistical Analysis

Statistical analyses were conducted using the Number Cruncher Statistical Systems 2007 (Keysville, UT, USE). One-way analysis of variance (ANOVA) was used to test the effect of one independent variable on an outcome variable. Since each independent variable (e.g., cell line, treatment, and culture condition) only affected two independent groups at a time in this study, the two-sample *t*-test, which is considered a special case of one-way ANOVA, was used to determine if there was a statistically significant difference between the means of two independent groups. A paired samples *t*-test was used to compare two means from the same tumor. Pearson’s correlation coefficient was used to describe the strength of linear correlation between two variables. A two-sided *p*-value of less than 0.05 is considered statistically significant.

## 5. Conclusions

Although it is well accepted that the absent HTATIP2 expression is associated with the increased susceptibility to tumorigenesis and enhanced tumor invasion and metastasis, the impact of HTATIP2 deficiency in tumor cells on tumor adaptation to hypoxia remains unclear. In this study we demonstrated that the absence of HTATIP2 expression in A549 human NSCLC cells resulted in accelerated tumor growth despite stalled tumor neovascularization and reduced tumor oxygenation, and lowered tumor sensitivity to sorafenib treatment. The tumor-promoting effect of the absent HTATIP2 expression was associated with HIF2α upregulation and enhancement of c-Myc oncogenic activity through the HIF2α regulated β-catenin/c-Myc/MCL-1 pathway. We showed that the HTATIP2-deficient A549 tumors were more susceptible to sorafenib-elicited EMT process in comparison to the A549 parental tumors. Results of the metabolomic analysis indicated that HTATIP2-deficient A549 tumors exhibited increased metabolic plasticity. Collectively, *the data suggest* that the absence of HTATIP2 expression promotes an aggressive tumor phenotype by enhancing the HIF2α-regulated β-catenin/c-Myc/MCL-1 signaling, increasing the susceptibility of tumors to sorafenib-activated EMT process, and improving tumor metabolic plasticity. Given the emerging evidence of the association of low HTATIP2 protein expression with high risk of metastasis and poor prognosis in NSCLC [81], by elucidating the distinct role of HTATIP2 in orchestrating tumor adaptation to hypoxic stress, the present study sets the groundwork for further attempts to identify new targets for therapeutic intervention in HTATIP2-deficient NSCLC.

## Figures and Tables

**Figure 1 cancers-12-01538-f001:**
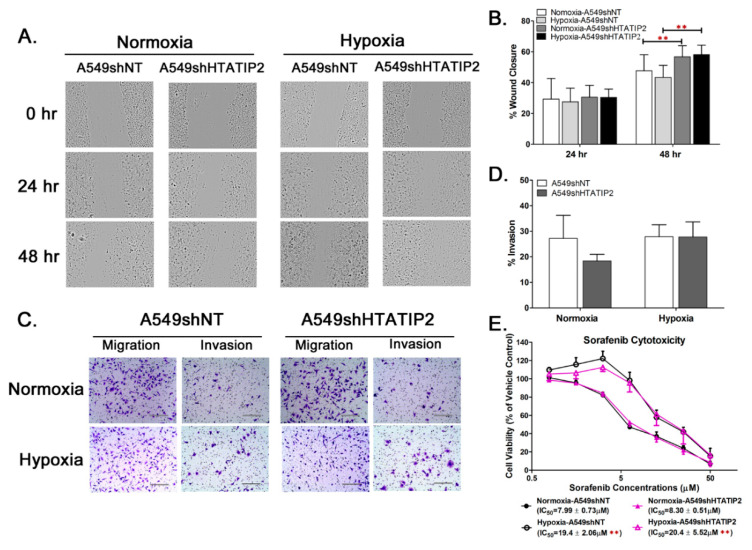
HTATIP2-knockdown exhibited increased migration potential and unchanged invasion potential and response to sorafenib treatment *in vitro* in A549 cells. (**A**). Representative micrographs of scratch wound closure kinetics of cultured A549shNT and A549shHTATIP2 cells cultured in serum-free DMEM/F12 medium under hypoxic and normoxic conditions. Original magnifications, × 100. (**B**). Quantification of relative wound area at 24 and 48 h after wounding of the cell monolayers, expressed as % wound closure, demonstrated the migration potential of the A549shHTATIP2 cells was significantly higher than that of the A549shNT cells at 48 h after wounding regardless of the culture conditions. Error bars represent the standard deviation (SD) of the mean from 8 independent experiments. Results are presented as mean ± SD. ** *p* < 0.01 compared with the counterpart A549shNT cells under the same culture condition using the two-sample *t*-test. (**C**). Representative micrographs from the transwell invasion assay performed by plating A549shNT and A549shHTATIP2 cells on uncoated and Matrigel-coated transwell membranes with 8-μm pore size and maintained for 14 h under hypoxic and normoxic conditions. Original magnifications, × 200. Scale bar, 200 μm. (**D**). No significant difference in the *in vitro* invasion potential was found between those two cell lines under hypoxic and normoxic conditions based on the percent invasion value, which was calculated as the percent of invaded cells relative to migrated cells. Error bars represent the SD of the mean from 3 independent experiments. Results are presented as mean ± SD. (**E**). Composite dose-effect curves for *in vitro* antiproliferative activity of sorafenib in A549shNT and A549shHTATIP2 cells. Error bars represent the inter-assay SD of the mean from 8 independent experiments. Results are presented as mean ± SD. ** *p* < 0.01 compared between normoxia and hypoxia conditions in the same cell line using the two-sample *t*-test.

**Figure 2 cancers-12-01538-f002:**
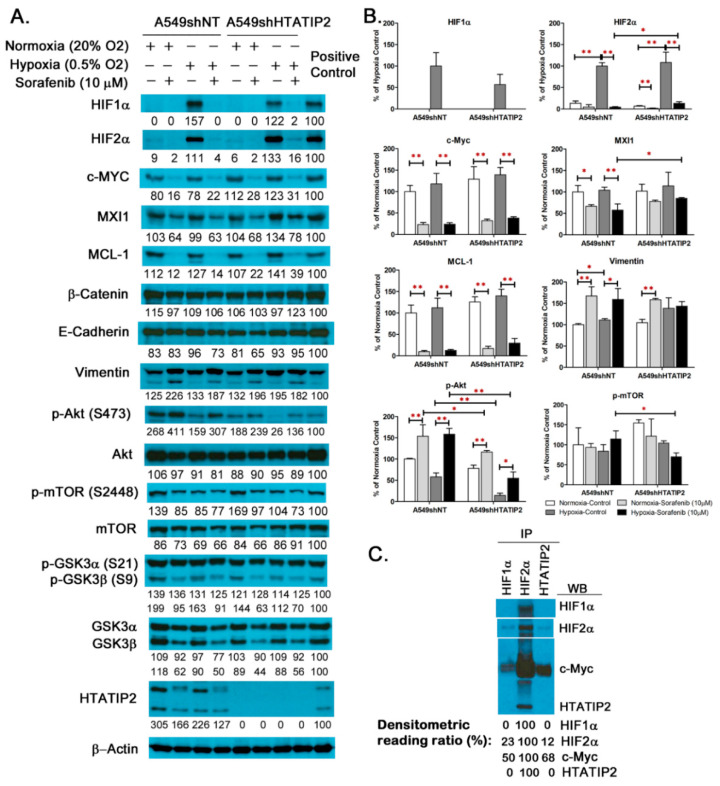
HTATIP2 knockdown potentially affected certain signaling pathways involved in tumor cell response to sorafenib treatment under hypoxic condition. (**A**). Representative Western blot images reflect the differential expression of certain proteins in A549shNT and A549shHTATIP2 cells cultured under hypoxic and normoxic conditions in the presence and absence of 10 μM sorafenib for 24 h. The positive control was used to compensate for both systematic and *random* errors from SDS-PAGE, membrane transfer, immunoblotting and chemiluminescence detection. The uncropped blots and molecular weight markers are shown in Appendix A. (**B**). Expression levels of selected proteins involved in cellular response to hypoxia and sorafenib treatment were determined by the quantification of Western blot images using the densitometric analysis. Relative immunoreactive band intensities are expressed as percent change over the average signal value in vehicle control A549shNT cultured under hypoxic (for HIF1α and HIF2α) or normoxic (for all other proteins tested) condition with normalization to the β-actin loading control and to the positive control. For phosphorylated proteins, results are expressed as the ratio of phosphorylated-to-total species relative to the vehicle control A549shNT cultured under normoxic condition. Results are presented as mean ± SD. SD is denoted by the error bars. * *p* < 0.05 and ** *p* < 0.01 compared between vehicle and sorafenib treatment in the same cell line under the same culture condition, or between A549shNT and A549shHTATIP2 cell lines receiving the same treatment and under the same culture condition, or between normoxic and hypoxic conditions in the same cell line with the same treatment using the two-sample *t*-test. (**C**) Western blot analysis of the immunoprecipitates of HIF1α, HIF2α, and HTATIP2 from A549 parental cells cultured under hypoxic condition for 24 h revealed interactions among HTATIP2, HIF2α and c-Myc and the absence of interaction between HTATIP2 and HIF1α.

**Figure 3 cancers-12-01538-f003:**
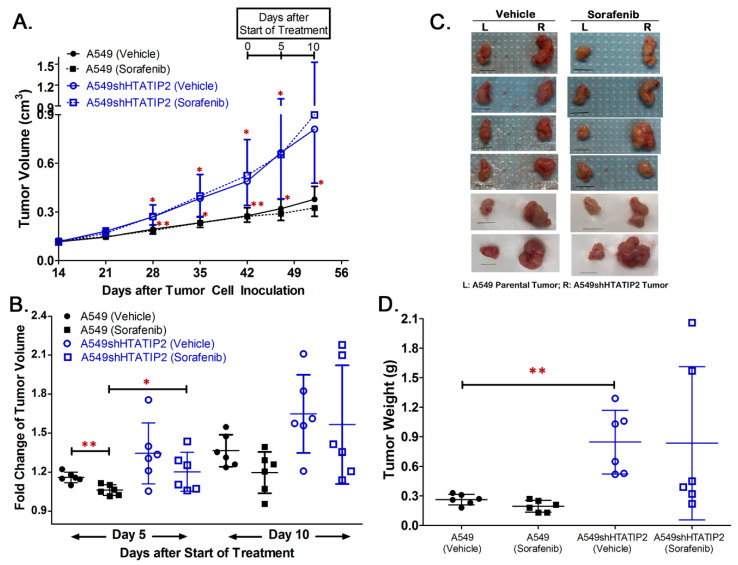
A549shHTATIP2 tumors exhibited rapid growth and reduced sensitivity to sorafenib treatment *in vivo*. Male athymic nude mice bearing both subcutaneous A549 parental (on the left flank) and A549shHTATIP2 (on the right flank) tumors received oral administration of either vehicle or 40 mg/kg/day of sorafenib for 10 consecutive days (*n* = 6 per group). (**A**). Comparison of tumor growth between A549 parental and A549shHTATIP2 tumors. Once daily sorafenib treatment at 40 mg/kg was started on Day 43 after tumor inoculation. (**B**). Fold change of tumor volume was compared between A549 and A549shHTATIP2 tumors and between vehicle and sorafenib treatment groups on Day 5 and Day 10 after the start of the treatment. Fold change of tumor volume was calculated as the ratio of tumor volume measured on Day 5 and Day 10 of the treatment period to that measured on Day 0 prior to the start of the treatment. (**C**). Subcutaneous tumors harvested from the vehicle and sorafenib treatment groups. Scale bar, 1 cm. (**D**). Comparison of tumor weights between A549 and A549shHTATIP2 tumors in the presence and absence of sorafenib treatment. Results are presented as mean ± SD. SD is denoted by the error bars. * *p* < 0.05 and ** *p* < 0.01 compared between A549 parental and A549shHTATIP2 tumors receiving the same treatment or between vehicle and sorafenib treatment in the same type of tumor using the two-sample *t*-test.

**Figure 4 cancers-12-01538-f004:**
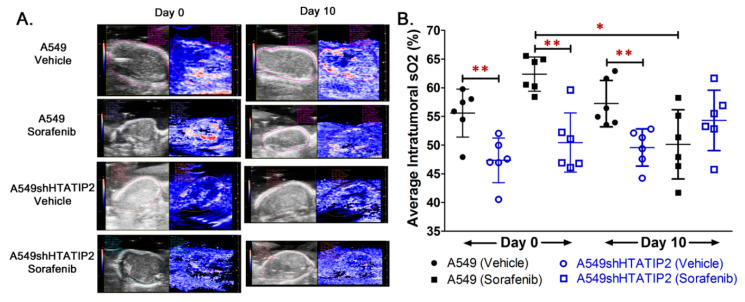
Oxygenation in vehicle-treated A549shHTATIP2 tumors was lower than that in their parental counterparts and was not affected by the sorafenib treatment. Real-time oxygen saturation (sO_2_) in A549 parental and A549shHTATIP2 tumors was evaluated using the Vevo^®^ LAZR photoacoustic imaging system on Day 0 before the start of the vehicle or sorafenib treatment (i.e., Day 42 after the subcutaneous tumor cell inoculation) and on Day 10 of the treatment period at 2 h after the last dose. (**A**). Representative photoacoustic oxygenation maps (Right) and ultrasound images (Left) taken on Day 0 and Day 10 for each study group. Individual oxyhemo images were shown side by side with the 2D B-model ultrasound image of the corresponding subcutaneous tumors. (**B**). Comparison of sO_2_ values between A549 parental and A549shHTATIP2 tumors in the same treatment group on the same day using the two-sample *t-*test with ** *p* < 0.01. Comparison between baseline and 10 days after the start of the same treatment in the same tumor type was performed using the paired sample *t*-test with * *p* < 0.05. Results are presented as mean ± SD. Error bars represent SD.

**Figure 5 cancers-12-01538-f005:**
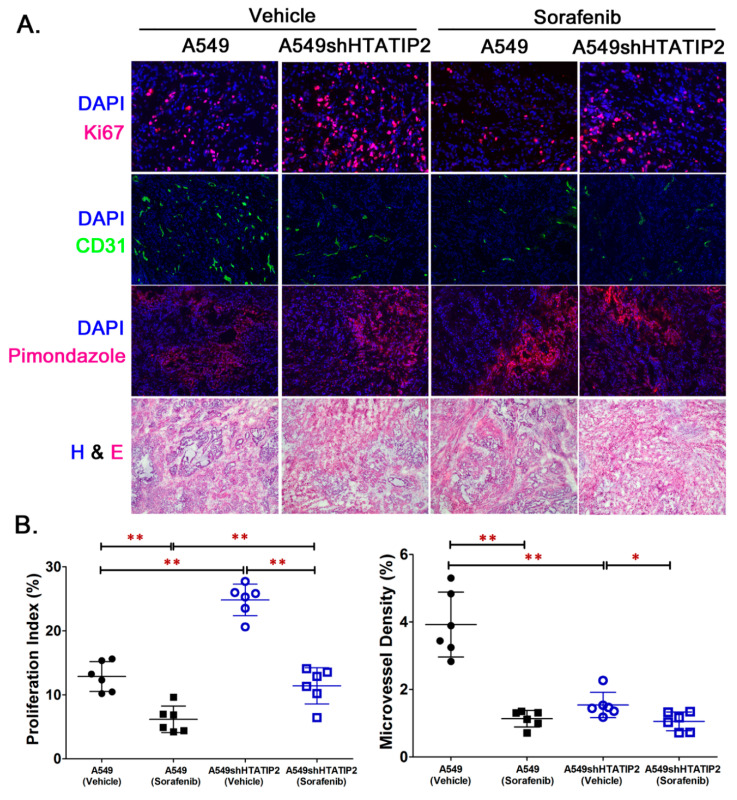
The aggressive phenotype of A549shHTATIP2 tumors was associated with increased number of proliferating cells and stalled tumor angiogenesis insensitive to sorafenib treatment. (**A**). Representative images for immunofluorescence staining for Ki67, CD31 and pimonidazole in tumor sections from individual groups. The nuclei were stained with 4’,6-diamidino-2-phenylindole (DAPI). Original magnifications, × 200. Immunofluorescence staining for pimonidazole indicated the existence of hypoxia in all tumor xenografts. The H&E staining was performed to assess tumor section integrity that might be affected during cryosectioning and fixation in 4% PFA. (**B**). Quantification of Ki67 and CD31 immunofluorescence staining showed increased number of proliferating cells and reduced micro-vessel density (MVD) in the vehicle-treated A549shHTATIP2 tumors. Sorafenib treatment significantly decreased the number of proliferating cells in both A549 and A549shHTATIP tumors and the MVD in A549 tumors, but had no effect on the MVD in A549shHTATIP2 tumors. Data are presented as mean ± SD. Error bars are SD. * *p* < 0.05 and ** *p* < 0.01 compared between A549 parental and A549shHTATIP2 tumors receiving the same treatment or between vehicle and sorafenib treatment in the same type of tumor using the two-sample *t*-test.

**Figure 6 cancers-12-01538-f006:**
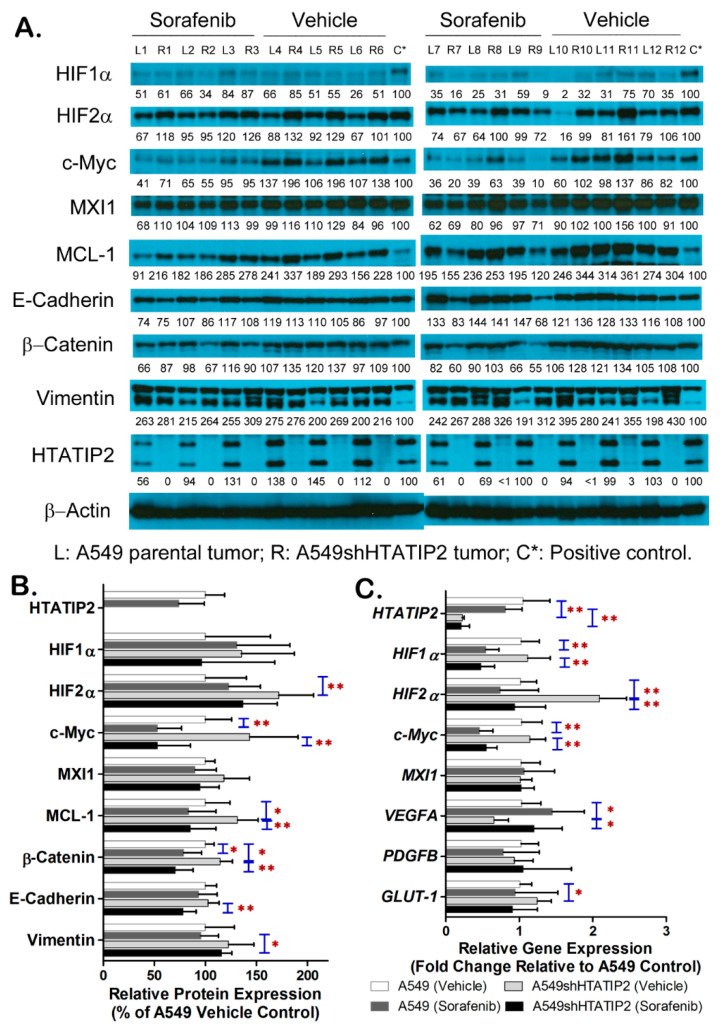
The aggressive phenotype of A549shHTATIP2 tumors was associated with the upregulated HIF2α expression, enhanced β-catenin/c-Myc/MCL-1 signaling and increased susceptibility of tumor cells to sorafenib-activated EMT process. (**A**). Western blot images of the expression of HIF1α, HIF2α, selected HIF-regulated proteins and EMT markers in individual tumor samples. The positive control was used to compensate for both systematic and random errors from SDS-PAGE, membrane transfer, immunoblotting and chemiluminescence detection. (**B**). Quantification of Western blot images using the densitometric analysis. Relative immunoreactive band intensities are expressed as percent change over the average signal value in vehicle control A549 parental tumors with normalization to the β-actin loading control and to the positive control. (**C**) Fold change of gene expression determined using the quantitative real-time PCR analysis and calculated using the 2^-ΔΔCt^ method. Data represent the *GAPDH*-normalized target gene expression level relative to that in the vehicle control A549 tumors which is considered 1. Data are presented as mean ± SD. Error bars are SD. * *p* < 0.05 and ** *p* < 0.01 compared between A549 parental and A549shHTATIP2 tumors receiving the same treatment or between vehicle and sorafenib treatment in the same type of tumor using the two-sample *t*-test.

**Table 1 cancers-12-01538-t001:** Metabolites with fold change of concentration greater than 2 in the A549shHTATIP2 tumors (*n* = 6) relative to the A549 parental tumors (*n* = 4) or with a *p* value less than 0.05.

Metabolites	Fold Increase ^a^	*p*-Value
L-Homocysteine	5.01	**0.007 ^c^**
L-Cysteine	2.9	0.104
Decanoyl-L-Carnitine	7.91	0.18
Lauroyl-L-Carnitine	2.98	0.057
Myristoyl-L-Carnitine	2.69	**0.01**
Stearoyl-L-Carnitine	2.18	0.062
Octanoyl-L-Carnitine	2.14	**0.02**
AICA-Riboside	2.07	0.085
**Metabolites**	**Fold Decrease ^b^**	***p*-Value**
Kynurenic acid (KYNA)	6.92	0.179
Malonyl CoA	6.70	0.145
ADP ribose	4.50	0.126
Coenzyme A	4.06	**0.006**
Glutathione	3.16	**0.003**
ATP	2.81	0.109
dGDP	2.56	0.195
NADP	2.56	**0.013**
ADP	2.54	0.162
Propionyl-CoA	2.50	**0.002**
Thiamine	2.43	**0.006**
N-Acetylornithine	2.33	**0.001**
Acetyl-CoA	2.30	**0.034**
β-Nicotinamide D-Ribonucleotide	2.19	**0.015**
Guanosine	2.12	**0.007**
Ethanolamine	2.02	**0.008**
Succinyl-CoA	1.99	**0.044**
Glutamine	1.94	**0.005**

Note: a: Fold increase is calculated as (Mean concentration in A549shHTATIP2 tumors)/(Mean concentration in A549 parental tumors). b: Fold decrease is calculated as (Mean concentration in A549 parental tumors}/(Mean concentration in A549shHTATIP2 tumors). c. The two-sample *t* test was used to compare means between two independent groups, and the statistically significant difference (*p* < 0.05) was marked with bold fonts.

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
