# Peer review of "Absence of HTATIP2 Expression in A549 Lung Adenocarcinoma Cells Promotes Tumor Plasticity in Response to Hypoxic Stress"

_cancers, 2020, doi:10.3390/cancers12061538_

Round 1
Reviewer 1 Report
The manuscript intended for Cancers aims to test the hypothesis that the tumor suppresor HTATIP2 plays a dominant role in downregulating malignant phenotypes associated with the progression of non-small cell lung cancer (NSCLC). In order to do so, the authors have gathered a considerable body of in vitro and in vivo evidence that HTATIP2, indeed, impedes the growth of NSCLC-derived cells and tumors, respectively. In addition, the effect of oxygen deprivation and pharmacological intervention using a broad kinase inhibitor and an epithelial-to-mesenchymal transition (EMT) inducer sorafenib have been investigated and the findings were correlated with altered changes in signal transduction and metabolic pathways. The authors have concluded that the association between HTATIP2, HIF2α, and c-Myc may represent an important safeguard mechanism against oncogenic phenotypes such as cancer cell motility, drug resistance, EMT, metabolic plasticity, and hypoxia. HTATIP2-deficient tumors were also characterized by activated β-catenin and MCL-1 signaling and reduced angiogenesis. The experimental data, which were obtained using a diverse methodological toolbox, fully support authors' claims. The manuscript is rigorously written and easy to read. However, more attention should be paid to microscopy image presentation, especially regarding sufficient pixel resolution. Also, calculations pertaining the metabolomic results could not be straightforwardly recapitulated (see minor points below).
1) The statement "The finding of significantly decreased E-cadherin (p < 0.01 compared with vehicle-treated A549shHTATIP2 tumors) and increased vimentin (p < 0.05 compared with sorafenib-treated A549 tumors) protein levels" (line 31) shoud mention the fact that both E-cadherin and vimentin expression changes were observed in A549shHTATIP2 tumors for better understanding.
2) Please change "acid" to "acids" (line 38).
3) Please replace "to treatment-activated" with "to sorafenib treatment-activated" (line 42).
4) There is a discontinuity in the text flow between line 58 and 59. Please fix.
5) Replace "sever" with "severe" (line 67).
6) MCL-1 could be added next to Bxl-xL as an anti-apoptotic protein example (line 88).
7) Please change "PRNA" to "RNA" (line 90), if this is the intended enzyme name.
8) Could the authors please provide cataglog number for the DMEM/F12 medium used (line 136)?
9) Could the authors please be more specific about the slight modification they applied to the wound-healing assay and by what means was the wound induced (line 152)?
10) Similarly, would it be possible to specify the slight modification used in the cell invasion assay (line 158)?
11) Please replace "was" with "were" (line 177).
12) Please change "has" to "have" (lines 186 and 633).
13) "promote" should be in singular form "promotes" (line 189).
14) The phrase "Each tumor was divided into sections for snap-frozen on dry ice" seems not to be gramatically correct (line 204).
15) "photoacoustic imaging (PAI)" could be shortened just to "PAI" (line 212) as the abbreviation was already defined above (line 196).
16) The link "http://rsb.info.nih.gov/ij/" does not work as it redirects to ScienceDirect (lines 231 and 240).
17) Please change "sampled" to "samples" (line 233).
18) Please change "and" to "or" (line 234).
19) Please replace "1965092" with "11965092", in case this is the correct catalog number for DMEM (line 235).
20) Please replace "hours, tumor" with "hours. Tumor" (line 236) to separate individual sentences from each other.
21) Please indicate in the methods section whether western blotted samples were prepared under reducing or non-reducing conditions and what amount of protein was loaded on gel.
22) There must be a grammar mistake in the sentence "individual metabolites concentrations were log-transformed and then subtracted the mean" (line 266) as the mathematical operation "and then substracted the mean" does not make sense.
23) Please either provide information on relative magnification or add scale bars, including length description in the respective figure legends, for Figure 3C and 4A.
24) Please describe better the migration result of Figure 1C in the text of the manuscript. Are there any significant differences? A quantification similar to that for the invasion assay presented in Figure 1D would be useful.
25) Please provide length description for scale bars presented in Figure 1C and Supplemental Figure S1B in the respective figure legends.
26) Please realign the graph legend of Figure 1E so that the left edge of the "(IC50=20.4 ± 5.52uM ++)" label exactly matches the left justification of all labels above this legend.
27) Please replace "HTATIP2-knockdown A549 cells exhibited increased migration potential and unchanged invasion potential and response to sorafenib treatment in vitro" (line 304) with "HTATIP2-knockdown exhibited increased migration potential and unchanged invasion potential and response to sorafenib treatment in vitro in A549 cells".
28) Please change "Figure 1A and 1B" to "Figure 2A and 2B" (lines 332, 338, and 342).
29) Please delete "and" (line 343) to make it more clear that only one condition is being compared (hypoxia + sorafenib).
30) The sentence "As shown in Figure 2C, HIF1α and HTATIP2 co-immunoprecipitated with HIF2α and so did HIF2α co-immunoprecipitate with antibodies recognizing HIF1α or HTATIP2" (line 353) could mention the fact that co-immunoprecipitation was carried under hypoxic condition.
31) It is not clear what does "positive control" refer to in Figure 2A, 5C, and Supplemental figure S2A. Could this information be appended to the respective figure legends and/or the Methods section?
32) Please provide p value confidence intervals for "*" and "**" and the test used to derive these parameters in the legend to Figure 2.
33) There is a conflict in the sentence "Comparison of the mean %sO2 values before and after sorafenib treatment indicated that the mean %sO2 value was significantly reduced by 20% (p < 0.05; paired samples t-test) in sorafenib-treated" (line 412) as statistically significant 20% decrease is actually achieved in sorafenib-treated parental A549 cells between Day 0 and 10. If there is indeed a significant difference between mean %sO2 values before and after sorafenib treatment in parental A549 cells, indicate this fact using asterisk(s) in Figure 4B and provide correct percentage change in the main text.
34) Please provide enlarged version of the photoacoustic oxygenation map and ultrasound images presented in Figure 4A as part of a new supplementary figure so that fine features can be seen in detail.
35) Please indicate in the legend to Figure 4A (line 443) which side (left or right) refers to photoacoustic oxygenation maps and which side corresponds to ultrasound imaging.
36) Please enlarge all images presented in Figure 5A to increase their resolution either by redistributing this panel to new figure(s) in the main text or by providing enlarged version of these images as part of a new supplementary figure so that readers can better appreciate fine morphological/fluorescence patterns of the stained tumor sections.
37) The pimonidazole result in Figure 5A seems to be vague as it is insufficiently described in the text. Does it actually indicate increased hypoxia in sorafenib-treated and A549shHTATIP2 tumors? A quantification similar to that displayed for Ki67 and CD31 staining presented in Figure 5B or a brief comment made in the main text as to why quantification was not possible could help to resolve this problem.
38) Similarly, it is not clear what the H&E staining result in Figure 5A suggests. Would it please be possible to include a brief comment in the figure legend and/or the main text on why the H&E staining is included or what features it describes? Any relevant structures such as blood vessels could perphaps be annotated.
39) The logic of the statement "demonstrating that A549shHTATIP2 tumors were less responsive to sorafenib treatment than A549 tumors" (line 487) is not clear as the beginning of that sentence "Overall, the Ki67 and CD31 immunofluorescent staining data were consistent with the relatively rapid growth and low oxygenation observed in the A549shHTATIP2 tumors" (line 485), comparing A549 and A549shHTATIP2 tumors, does not give any reason to infer conclusions regarding sorafenib sensitivity. Please rephrase or separate text into two sentences. In addition, A549shHTATIP2 tumors were seemingly more sensitive to sorafenib treatment than A549 tumors in terms of the estimated proliferation index (Figure 5B top). Could the authors make a brief comment in the text of the manuscript as to how this discrepancy can be reconciled with the tumor volume and weight data (Figure 3)?
40) Please replace "that their" with "that in their" (line 522).
41) The percentage changes for CoA, acetyl-CoA, succinyl-CoA, and propionyl-CoA (line 552), glutamine and β-nicotinamide D-ribonucleotide (line 553), and guanosine and glutathione (line 554) do not exactly correspond to values listed in Table 1. For example, for CoA with a Fold Decrease of 4.06 (Table 1) one would expect this to result in a 100*(1 - 1/4.06) = 75% change, which is a different value from 48% provided in the text (line 552).
42) The Fold Increase and Fold Decrease values listed in Table 1 do not exactly correspond to values calculated from mean concentrations listed in Supplemental Table S3. For example, if L-homocysteine Fold Increase = [Mean concentration in A549shHTATIP2 tumors]/[Mean concentration in A549 parental tumors] than one would expect 5.01 (Table 1) = 16.769 / 3.525 (Supplemental Table S3, line 186, M259). However, this calculation yields Fold Increase of only 4.75.
43) There is an extra period (line 565).
44) Could the authors add a brief assessment of the metastatic potential of HTATIP2-deficient tumors and/or whether they plan to experimentally follow up on it into the Discussion session?
45) Please replace "(Figure 5C – 5E)" with "(Figure 5C and 5D)" (lines 638 and 658).
46) Please define NSCLC abbreviation (line 641). It may also be beneficial to recognize A549 cells as NSCLC earlier such as in the Introduction section.
47) The sentence "Moreover, upon the treatment with sorafenib, the β-catenin and E-cadherin protein levels in A549shHTATIP2 tumors were significantly reduced (p < 0.01 for both as compared with vehicle-treated A549shHTATIP2 tumors) while the vimentin protein level was significantly elevated (p < 0.05 compared with sorafenib-treated A549 parental tumors) (Figure 5C – 5E)" (line 654) is confusing since, in the first part, expression between sorafenib-treatead and untreated A549shHTATIP2 tumors is being compared, whereas in the second part, expression is being compared between sorafenib-treated parental and A549shHTATIP2 tumors. Would it please possible to separate this text into two sentences?
48) Please change "effecter" to "effector" (line 658).
49) Please replace "(Table 1)" with "(Table 1) (Figure 5E) (line 680).
50) Could the Supplemental Materials and Methods section content be moved and merged with the Methods section in the main text?
51) It is not clear what chemical agent the authors mean by "phenocumorin" in the Supplemental Materials and Methods. Please revise.
52) Please replace "SJ-MES-1-shHTATIP2" with "SK-MES-1-shHTATIP2" in the legend to Supplemental Figure S1.
Author Response
Reviewer 1 comments:
Comments and Suggestions for Authors
The manuscript intended for Cancers aims to test the hypothesis that the tumor suppressor HTATIP2 plays a dominant role in downregulating malignant phenotypes associated with the progression of non-small cell lung cancer (NSCLC). In order to do so, the authors have gathered a considerable body of in vitro and in vivo evidence that HTATIP2, indeed, impedes the growth of NSCLC-derived cells and tumors, respectively. In addition, the effect of oxygen deprivation and pharmacological intervention using a broad kinase inhibitor and an epithelial-to-mesenchymal transition (EMT) inducer sorafenib have been investigated and the findings were correlated with altered changes in signal transduction and metabolic pathways. The authors have concluded that the association between HTATIP2, HIF2α, and c-Myc may represent an important safeguard mechanism against oncogenic phenotypes such as cancer cell motility, drug resistance, EMT, metabolic plasticity, and hypoxia. HTATIP2-deficient tumors were also characterized by activated β-catenin and MCL-1 signaling and reduced angiogenesis. The experimental data, which were obtained using a diverse methodological toolbox, fully support authors' claims. The manuscript is rigorously written and easy to read. However, more attention should be paid to microscopy image presentation, especially regarding sufficient pixel resolution. Also, calculations pertaining the metabolomic results could not be straightforwardly recapitulated (see minor points below).
Reply: We appreciate the reviewer’s comments on the microscopy images. We managed to enlarge individual images without losing the resolution of 300 dpi. As a result, Figure 5C, 5D and 5E were removed from Figure 5 and presented as Figure 6.
1) The statement "The finding of significantly decreased E-cadherin (p < 0.01 compared with vehicle-treated A549shHTATIP2 tumors) and increased vimentin (p < 0.05 compared with sorafenib-treated A549 tumors) protein levels" (line 31) should mention the fact that both E-cadherin and vimentin expression changes were observed in A549shHTATIP2 tumors for better understanding.
Reply: The sentence has been modified with the addition of “in A549shHTATIP2 tumors” (Line 33).
2) Please change "acid" to "acids" (line 38).
Reply: Correction has been made accordingly.
3) Please replace "to treatment-activated" with "to sorafenib treatment-activated" (line 42).
Reply: Correction has been made accordingly.
4) There is a discontinuity in the text flow between line 58 and 59. Please fix.
Reply: The discontinuity between line 58 and 59 has been fixed.
5) Replace "sever" with "severe" (line 67).
Reply: The typographical error has been fixed.
6) MCL-1 could be added next to Bxl-xL as an anti-apoptotic protein example (line 88).
Reply: Thanks for the suggestion. MCL-1 has been added after Bxl-xL.
7) Please change "PRNA" to "RNA" (line 90), if this is the intended enzyme name.
Reply: “PRNA” has been changed to “RNA”.
8) Could the authors please provide catalog number for the DMEM/F12 medium used (line 136)?
Reply: The following information has been added “(CorningTM 10-092-CV. Fisher Scientific, Waltham, MA)” (Line 136 – 137)
9) Could the authors please be more specific about the slight modification they applied to the wound-healing assay and by what means was the wound induced (line 152)?
Reply: The slight modification was that the hypoxic culture condition was used in this study as mentioned in Line 158, while the studies cited were not done under the hypoxia culture condition. The following sentence has been added to indicate how wound areas on cell monolayers were created: “The wound areas on cell monolayers were created using a sterile 10μl (P10) pipette tip.” (Line 159 – 160)
10) Similarly, would it be possible to specify the slight modification used in the cell invasion assay (line 158)?
Reply: The slight modification was that the hypoxic culture condition was used in this study as mentioned in Line 163, while the study cited did not include the hypoxic culture condition.
11) Please replace "was" with "were" (line 177).
Reply: Thanks for spotting the grammatical error. The correction has been made to the sentence.
12) Please change "has" to "have" (lines 186 and 633).
Reply: Corrections have been made accordingly.
13) "promote" should be in singular form "promotes" (line 189).
Reply: The correction has been made accordingly.
14) The phrase "Each tumor was divided into sections for snap-frozen on dry ice" seems not to be gramatically correct (line 204).
Reply: The sentence has been changed to “Each tumor was divided into several pieces for snap-freezing on dry ice or preserving in TRIzol reagent” (Line 213)
15) "photoacoustic imaging (PAI)" could be shortened just to "PAI" (line 212) as the abbreviation was already defined above (line 196).
Reply: The correction has been made accordingly.
16) The link "http://rsb.info.nih.gov/ij/" does not work as it redirects to ScienceDirect (lines 231 and 240).
Reply: We apologize for the outdated link. The correct link (https://imagej.nih.gov/ij/) has been added to the text (Line 239 and 252)
17) Please change "sampled" to "samples" (line 233).
Reply: The correction has been made.
18) Please change "and" to "or" (line 234).
Reply: The correction has been made.
19) Please replace "1965092" with "11965092", in case this is the correct catalog number for DMEM (line 235).
Reply: The correction has been made.
20) Please replace "hours, tumor" with "hours. Tumor" (line 236) to separate individual sentences from each other.
Reply: The correction has been made.
21) Please indicate in the methods section whether western blotted samples were prepared under reducing or non-reducing conditions and what amount of protein was loaded on gel.
Reply: The following information has been added to the “2.12. Semi-Quantitative Western Blot Analysis” sub-section to address the reviewer’s concerns: “Cell lysates and tumor tissue homogenates were reduced and denatured by boiling the samples in sample buffer containing dithiothreitol (DTT) and sodium dodecyl sulfate (SDS) at 100˚C for 5 minutes before loaded onto SDS-polyacrylamide gel electrophoresis (PAGE) gels as described previously. The amount of total protein loaded per lane was about 20 and 75 μg for cell lysates and tumor tissue homogenates, respectively.” (Line 245 – 250)
22) There must be a grammar mistake in the sentence "individual metabolites concentrations were log-transformed and then subtracted the mean" (line 266) as the mathematical operation "and then substracted the mean" does not make sense.
Reply: We appreciate the reviewer’s comment. The statement has been corrected as follows: “To meet the normality assumption, individual metabolite concentrations were log‐transformed and then autoscaled (mean‐centered and divided by the standard deviation of each metabolite).” (Line 290)
23) Please either provide information on relative magnification or add scale bars, including length description in the respective figure legends, for Figure 3C and 4A.
Reply: The scale bars have been added to Figure 3C. We could not figure out how to add scale bars to the photoacoustic images and B-mode ultrasound images in Figure 4A using the software (Vevo Lab) that comes with the Vevo LAZR-X PAI system.
24) Please describe better the migration result of Figure 1C in the text of the manuscript. Are there any significant differences? A quantification similar to that for the invasion assay presented in Figure 1D would be useful.
Reply: The following sentence “Results of the wound-healing assay indicated that the percent wound closure value of A549shNT cells was significantly lower than that of A549shHTATIP2 cells at 48 hr after wounding under both normoxic and hypoxic conditions (N = 8; p < 0.01 for both conditions) (Figure 1A and 1B),” has been changed to “Results of the wound-healing assay indicated that the mean percent wound closure values of A549shNT cells were significantly lower than that of A549shHTATIP2 cells at 48 hr after wounding under both normoxic (16% decrease) and hypoxic (26% decrease) conditions (N = 8; p < 0.01 for both conditions) (Figure 1A and 1B), indicating that A549shHTATIP2 cells possess greater migration potential than A549shNT cells under the same culture condition.” (Line 311 – 316). The following sentence “Results of the 14-hr transwell assay showed no significant difference in the invasion potential between A549shNT and A549shHTATIP2 cells under normoxic and hypoxic conditions (N = 3; p > 0.05 for both conditions) (Figure 1C and 1D)” has been changed to “Results of the 14-hr transwell assay showed no significant difference in the invasion potential between A549shNT and A549shHTATIP2 cells under normoxic (27% ± 9% vs 18% ± 3%) and hypoxic (28% ± 5% vs 28% ± 6%) conditions (N = 3; p > 0.05 for both conditions) (Figure 1C and 1D)” to include the quantification information. (Line 316-318).
25) Please provide length description for scale bars presented in Figure 1C and Supplemental Figure S1B in the respective figure legends.
Reply: The following information has been added to figure legends of Figure 1C and Figure S1B: “Scale bar, 200 μm”.
26) Please realign the graph legend of Figure 1E so that the left edge of the "(IC50=20.4 ± 5.52uM ++)" label exactly matches the left justification of all labels above this legend.
Reply: Due to the limited space, we had to move the graph legend of Figure 1E to the bottom of the figure so we can increase the font size. Although it does not match the left justification of all labels above, at least we make it easier to read.
27) Please replace "HTATIP2-knockdown A549 cells exhibited increased migration potential and unchanged invasion potential and response to sorafenib treatment in vitro" (line 304) with "HTATIP2-knockdown exhibited increased migration potential and unchanged invasion potential and response to sorafenib treatment in vitro in A549 cells".
Reply: The first sentence of the figure legend of Figure 1 has been changed to “HTATIP2-knockdown exhibited increased migration potential and unchanged invasion potential and response to sorafenib treatment in vitro in A549 cells” as per the reviewer’s suggestion.
28) Please change "Figure 1A and 1B" to "Figure 2A and 2B" (lines 332, 338, and 342).
Reply: "Figure 1A and 1B" has been changed to "Figure 2A and 2B". We apologize for the confusion.
29) Please delete "and" (line 343) to make it more clear that only one condition is being compared (hypoxia + sorafenib).
Reply: The word “and” has been deleted. It was indeed that only one source of variation was investigated at a time in this study.
30) The sentence "As shown in Figure 2C, HIF1α and HTATIP2 co-immunoprecipitated with HIF2α and so did HIF2α co-immunoprecipitate with antibodies recognizing HIF1α or HTATIP2" (line 353) could mention the fact that co-immunoprecipitation was carried under hypoxic condition.
Reply: The sentence has been changed to “Results of the co-immunoprecipitation assay using whole cell extracts from A549 cells cultured under hypoxia condition showed that … (Figure 2C)” (Line 384).
31) It is not clear what does "positive control" refer to in Figure 2A, 5C, and Supplemental figure S2A. Could this information be appended to the respective figure legends and/or the Methods section?
Reply: The following statement has been added to the figure legends of Figure 2, Figure 6 and Supplemental Figure S3: “The positive control was used to compensate for both systematic and random errors from SDS-PAGE, membrane transfer, immunoblotting and chemiluminescence detection”.
32) Please provide p value confidence intervals for "*" and "**" and the test used to derive these parameters in the legend to Figure 2.
Repy: The following sentence has been added to the legend to Figure 2: “Results are presented as mean ± SD. SD is denoted by the error bars. *p < 0.05 and **p < 0.01 using the two-sample t test.”
33) There is a conflict in the sentence "Comparison of the mean %sO2 values before and after sorafenib treatment indicated that the mean %sO2 value was significantly reduced by 20% (p < 0.05; paired samples t-test) in sorafenib-treated" (line 412) as statistically significant 20% decrease is actually achieved in sorafenib-treated parental A549 cells between Day 0 and 10. If there is indeed a significant difference between mean %sO2 values before and after sorafenib treatment in parental A549 cells, indicate this fact using asterisk(s) in Figure 4B and provide correct percentage change in the main text.
Reply: The sentence has been changed to “Comparison of the mean %sO2 values before and after sorafenib treatment indicated that sorafenib treatment significantly reduced the mean %sO2 value in A549 tumors by 20% (p < 0.05; paired samples t-test), but had no significant effect on the mean %sO2 value in A549shHTATIP2 tumors (p > 0.05) (Figure 4B).” (Line 454 – 455).
34) Please provide enlarged version of the photoacoustic oxygenation map and ultrasound images presented in Figure 4A as part of a new supplementary figure so that fine features can be seen in detail.
Replay: Enlarged representative PAI and ultrasound images have been included as the new Supplemental Figure S2.
35) Please indicate in the legend to Figure 4A (line 443) which side (left or right) refers to photoacoustic oxygenation maps and which side corresponds to ultrasound imaging.
Reply: The sentence in the legend to Figure 4A has been changed to “Representative photoacoustic oxygenation maps (Right) and ultrasound images (Left) …”
36) Please enlarge all images presented in Figure 5A to increase their resolution either by redistributing this panel to new figure(s) in the main text or by providing enlarged version of these images as part of a new supplementary figure so that readers can better appreciate fine morphological/fluorescence patterns of the stained tumor sections.
Reply: We were able to enlarge all images presented in Figure5A after moving Figure 5C-5E to Figure 6.
37) The pimonidazole result in Figure 5A seems to be vague as it is insufficiently described in the text. Does it actually indicate increased hypoxia in sorafenib-treated and A549shHTATIP2 tumors? A quantification similar to that displayed for Ki67 and CD31 staining presented in Figure 5B or a brief comment made in the main text as to why quantification was not possible could help to resolve this problem.
Reply: The following sentence has been added to explain why quantitative evaluation of pimonidazole staining could not be provided in this study: “Quantitative evaluation of pimonidazole stained tumor sections was not feasible due to the considerable difference in pimonidazole staining pattern among tumors.” (Line 548 – 550).
38) Similarly, it is not clear what the H&E staining result in Figure 5A suggests. Would it please be possible to include a brief comment in the figure legend and/or the main text on why the H&E staining is included or what features it describes? Any relevant structures such as blood vessels could perphaps be annotated.
Reply: In our lab, the H&E staining was routinely performed along with immunofluorescence staining and immunohistochemistry for the purpose of inspection of tumor section integrity that might be affected during cryosectioning and fixation in 4% PFA. The following statement has been added to the legend to Figure 5A to explain why the H&E staining result was included: “The H&E staining was performed to assess tumor section integrity that might be affected during cryosectioning and fixation in 4% PFA.”
39) The logic of the statement "demonstrating that A549shHTATIP2 tumors were less responsive to sorafenib treatment than A549 tumors" (line 487) is not clear as the beginning of that sentence "Overall, the Ki67 and CD31 immunofluorescent staining data were consistent with the relatively rapid growth and low oxygenation observed in the A549shHTATIP2 tumors" (line 485), comparing A549 and A549shHTATIP2 tumors, does not give any reason to infer conclusions regarding sorafenib sensitivity. Please rephrase or separate text into two sentences. In addition, A549shHTATIP2 tumors were seemingly more sensitive to sorafenib treatment than A549 tumors in terms of the estimated proliferation index (Figure 5B top). Could the authors make a brief comment in the text of the manuscript as to how this discrepancy can be reconciled with the tumor volume and weight data (Figure 3)?
Reply: We have modified the statement in the text as follows “Overall, the Ki67 and CD31 immunofluorescent staining data obtained from the vehicle control groups were consistent with the relatively rapid growth and low oxygenation observed in the A549shHTATIP2 tumors. Although the inhibitory effect of short-term sorafenib treatment on cell proliferation in the A549shHTATIP2 tumors was comparable to that in the A549 tumors, the antiangiogenic effect of sorafenib was absent in the A549shHTATIP2 tumors. This observation raises a question: how is the growth of A549shHTATIP2 tumors sustained with limited blood supply?” (Line 552 – 556)
40) Please replace "that their" with "that in their" (line 522).
Reply: We have replaced “that their” with “that in their”.
41) The percentage changes for CoA, acetyl-CoA, succinyl-CoA, and propionyl-CoA (line 552), glutamine and β-nicotinamide D-ribonucleotide (line 553), and guanosine and glutathione (line 554) do not exactly correspond to values listed in Table 1. For example, for CoA with a Fold Decrease of 4.06 (Table 1) one would expect this to result in a 100*(1 - 1/4.06) = 75% change, which is a different value from 48% provided in the text (line 552).
Reply: We have recalculated the percent decrease for those metabolites and provided the correct data in the text as follows: “In particular, the mean levels of NADP, CoA, acetyl-CoA and succinyl-CoA in A549shHTATIP2 tumors were significantly decreased by 61% (p < 0.05), 75% (p < 0.01), 57% (p < 0.01) and 50% (p < 0.01), respectively, and those of propionyl-CoA and ethanolamine by 60% and 51%, respectively (p < 0.01 for both), and that of glutamine by 48% (p < 0.01), and those of β-nicotinamide D-ribonucleotide and guanosine by 54% (p < 0.05) and 53% (p < 0.01), respectively, and those of glutathione and thiamine by 68% and 59%, respectively (p < 0.01 for both) (Table 1).” (Line 620 – 624)
42) The Fold Increase and Fold Decrease values listed in Table 1 do not exactly correspond to values calculated from mean concentrations listed in Supplemental Table S3. For example, if L-homocysteine Fold Increase = [Mean concentration in A549shHTATIP2 tumors]/[Mean concentration in A549 parental tumors] than one would expect 5.01 (Table 1) = 16.769 / 3.525 (Supplemental Table S3, line 186, M259). However, this calculation yields Fold Increase of only 4.75.
Reply: Thank you very much for catching our mistakes. The data presented in Table 1 were based on tumor weight normalized metabolite concentration, whereas those presented in Supplemental Table S3 were metabolite concentration in tumor homogenates. What was done in the sample preparation for metabolomic analysis was that the same volume of deionized water was added to the tumor sample to prepare tumor tissue homogenate. The resultant metabolite concentration was then normalized with the weight of the tumor sample used in the preparation of the tumor homogenate. For the data analysis, the tumor weight normalized metabolite concentrations (pmol/mg tissue) should be used. We have replaced the raw concentration data with the tumor weight normalized data in Supplemental Table S3
43) There is an extra period (line 565).
Reply: The extra period has been deleted.
44) Could the authors add a brief assessment of the metastatic potential of HTATIP2-deficient tumors and/or whether they plan to experimentally follow up on it into the Discussion session?
Reply: As per the reviewer’s suggestion, we have added the following sentences to the Discussion section (Line 768-786): “Upon the treatment with sorafenib, the β-catenin protein expression level decreased significantly in both A549 parental and A549shHTATIP2 tumors, coinciding with the significant decrease in c-Myc mRNA and protein levels (Figure 6). Since β-catenin is associated with c-Myc by functioning as a transcription factor to activate c-Myc [69], this result suggests that short-term sorafenib treatment inhibits the c-Myc-stimulated cell proliferation by decreasing the activity of β-catenin in both A549 and A549shHTATIP2 tumors. However, unlike the A549 tumors, sorafenib treatment resulted in not only a significantly decreased β-catenin protein level but also a significantly decreased E-cadherin protein level in A549shHTATIP2 tumors (p < 0.01 for both) as compared with the vehicle-treated A549shHTATIP2 tumors. It is speculated that the decreased b-catenin levels in A549shHTATIP tumor tissue homogenates is in part attributable to the decreased membranous and cytoplasmic b-catenin levels associated with the liberation of b-catenin from the cytoplasmic tail of E-cadherin [75] followed by the nuclear translocation of b-catenin that potentially facilitates the transcriptional changes associated with tumor metastasis [76,77]. In addition, the observation of the significant increase in vimentin protein expression in sorafenib-treated A549shHTATIP2 tumors compared to the sorafenib-treated A549 tumors (p < 0.05) (Figure 6A and 6B) reinforced our belief that the absence of HTATIP2 expression increases the susceptibility of A549 tumors to sorafenib-activated EMT process. In the future, it will be interesting to explore the metastasis signaling cascade associated with the adaptation of A549shHTATIP2 tumors to long-term sorafenib treatment. (Line 734-752).
45) Please replace "(Figure 5C – 5E)" with "(Figure 5C and 5D)" (lines 638 and 658).
Reply: “Figure 5C-5E” has been replaced with “Figure 6A and 6B”.
46) Please define NSCLC abbreviation (line 641). It may also be beneficial to recognize A549 cells as NSCLC earlier such as in the Introduction section.
Reply: The last sentence in the Introduction section has been changed to “In the present study, we elucidated a novel mechanism … in a murine xenograft model of A549 human lung adenocarcinoma, which represents the most common subtype of non-small cell lung carcinoma (NSCLC).” (Line 145 – 146)
47) The sentence "Moreover, upon the treatment with sorafenib, the β-catenin and E-cadherin protein levels in A549shHTATIP2 tumors were significantly reduced (p < 0.01 for both as compared with vehicle-treated A549shHTATIP2 tumors) while the vimentin protein level was significantly elevated (p < 0.05 compared with sorafenib-treated A549 parental tumors) (Figure 5C – 5E)" (line 654) is confusing since, in the first part, expression between sorafenib-treatead and untreated A549shHTATIP2 tumors is being compared, whereas in the second part, expression is being compared between sorafenib-treated parental and A549shHTATIP2 tumors. Would it please possible to separate this text into two sentences?
Reply: As per the reviewer’s suggestion, we have changed the statement to “… the expression of E-cadherin in sorafenib-treated A549shHTATIP2 tumors was significantly decreased by 24% compared to the vehicle-treated A549shHTATIP2 tumors (p < 0.01), implicating that sorafenib treatment significantly disrupts intercellular contacts in the A549shHTATIP2 tumors. The expression of vimentin in sorafenib-treated A549shHTATIP2 tumors was significantly increased by 21% compared to their sorafenib-treated parental counterparts (p < 0.05) (Figure 6A and 6B), suggesting that the absence of HTATIP2 expression increases the susceptibility of A549 tumors to sorafenib-activated EMT process.” (Line 580 – 583)
48) Please change "effecter" to "effector" (line 658).
Reply: We have changed “effecter” to “effector”.
49) Please replace "(Table 1)" with "(Table 1) (Figure 5E) (line 680).
Reply: We have replaced “Table 1” with “Table 1 and Figure 6C” as the previous Figure 5C, 5D and 5E have changed to Fiure 6A, 6B and 6C.
50) Could the Supplemental Materials and Methods section content be moved and merged with the Methods section in the main text?
Reply: We have moved the supplemental materials and method section to the main text.
51) It is not clear what chemical agent the authors mean by "phenocumorin" in the Supplemental Materials and Methods. Please revise.
Reply: We apologize for the typo. It should be phenylcoumarin.
52) Please replace "SJ-MES-1-shHTATIP2" with "SK-MES-1-shHTATIP2" in the legend to Supplemental Figure S1.
Reply: We have replaced "SJ-MES-1-shHTATIP2" with "SK-MES-1-shHTATIP2" in the legend to Supplemental Figure S1.
We are very thankful to the reviewer for his/her time and patience and all the insightful comments.

Reviewer 2 Report
This is a valuable contribution to the field - original and novel findings backed up by clever experimentation and cellular manipulations.
My big issue is with the data analysis, and whether many of the purported differences are in fact significant. Many of the differences reported are subtle, and rely on P<0.05 to gain attention. Thus, it is very important that the correct statistical techniques are used.
However, in multiple places (starting with Figure 1) there experiments with more than 2 groups appear to have been analysed with multiple pairwise t-tests, rather than post-hoc tests following ANOVA. This commits the multiple comparisons error and inflates Type I error.
Also, some points are unclear as to what has been done at all. E.g., Fig 2 has no explanation of asterisks at all. In Fig. 3A, it is unclear what kind of analysis has lead to the between group/same day P values. All this needs a thorough reworking.
Author Response
Reviewer 2 comments:
This is a valuable contribution to the field - original and novel findings backed up by clever experimentation and cellular manipulations.
My big issue is with the data analysis, and whether many of the purported differences are in fact significant. Many of the differences reported are subtle, and rely on P<0.05 to gain attention. Thus, it is very important that the correct statistical techniques are used.
However, in multiple places (starting with Figure 1) there experiments with more than 2 groups appear to have been analysed with multiple pairwise t-tests, rather than post-hoc tests following ANOVA. This commits the multiple comparisons error and inflates Type I error.
Also, some points are unclear as to what has been done at all. E.g., Fig 2 has no explanation of asterisks at all. In Fig. 3A, it is unclear what kind of analysis has lead to the between group/same day P values. All this needs a thorough reworking.
Reply: We appreciate the reviewer’s comments on the statistical analysis used in this study. Here is the reason that the two-sample t-test was used throughout this study. Given the limited sample size in this study, we were not able to use the two-way analysis of variance (ANOVA) for statistical analysis. With one-way ANOVA, only one source of variation, or independent variable, can be investigated at a time [Ref: Wayne W. Daniel. Biostatistics: A Foundation for Analysis in the Health Sciences. 7th Edition. Page 295 – 299]. In our case, there were 3 different sources of variation present in most of the in vitro experiments: (1) cell line (A549shNT vs A549shHTATIP2), (2) culture condition (normoxia vs hypoxia), and (3) treatment (vehicle vs sorafenib). Since there were only two independent groups in a situation in which one source of variation was present, the two-sample t-test was deemed to be appropriate. Likewise, in the in vivo study, there were two different sources of variation, i.e., cell line (A549 parental vs A549shHTATIP2) and treatment (vehicle vs sorafenib). The two-sample t-test was used to compare either between A549 and A549shHTATIP2 tumors receiving the same treatment or between vehicle control and sorafenib treatment in the same type of xenograft tumors. To clarify why the two-sample t-test was used in this study, we have added the following statement under the “2.16. Statistical Analyses” sub-section: “One-way analysis of variance (ANOVA) was used to test the effect of one independent variable on an outcome variable. Since each independent variable (e.g., cell line, treatment, and culture condition) only affected two independent groups at a time in this study, the two-sample t-test, which is considered a special case of one-way ANOVA, was used to determine if there was a statistically significant difference between the means of two independent groups.” (Line 293-297)
We have gone through the legends to individual figures to make sure that the information of statistical analysis was clearly provided as follows:
Figure 1. (A) **p < 0.01 compared with the counterpart A549shNT cells under the same culture condition using the two-sample t-test. (E) **p < 0.01 compared between normoxia and hypoxia conditions in the same cell line using the two-sample t-test.
Figure 2. (B) *p < 0.05 and **p < 0.01 compared between vehicle and sorafenib treatment in the same cell line under the same culture condition, or between A549shNT and A549shHTATIP2 cell lines receiving the same treatment and under the same culture condition, or between normoxic and hypoxic conditions in the same cell line with the same treatment using the two-sample t-test.
Figure 3. (D) *p < 0.05 and **p < 0.01 compared between A549 parental and A549shHTATIP2 tumors receiving the same treatment or between vehicle and sorafenib treatment in the same type of tumor using the two-sample t-test.
Figure 4. (B) Comparison of sO2 values between A549 parental and A549shHTATIP2 tumors in the same treatment group on the same day using the two-sample t-test with **p < 0.01. Comparison between baseline and 10 days after the start of the same treatment in the same tumor type was performed using the paired sample t-test with *p < 0.05.
Figure 5. (B) *p < 0.05 and **p < 0.01 compared between A549 parental and A549shHTATIP2 tumors receiving the same treatment or between vehicle and sorafenib treatment in the same type of tumor using the two-sample t-test.
Figure 6. (C) *p < 0.05 and **p < 0.01 compared between A549 parental and A549shHTATIP2 tumors receiving the same treatment or between vehicle and sorafenib treatment in the same type of tumor using the two-sample t-test.

Reviewer 3 Report
In this manuscript titled “Absence of HTATIP2 Expression in A549 Lung Adenocarcinoma Cells Promotes Tumor Plasticity in Response to Hypoxic Stress”, the authors helped delineate the mechanism by which the absent expression of HTATIP2 contributes to tumor adaptation to hypoxia and the tumor aggressiveness. The report has successfully clarified more about the role of HTATIP as a tumor suppressor. In general, the data are robust and the manuscript is clearly written. The reviewer recommends acceptance of the manuscript for publication after addressing the minor issues below:
- Figure 2: Please indicates the P values of those * and ** in the figure legend.
- Line 196-198: Measurement of intratumoral oxygen saturation was performed at 2 hours after the last dose of sorafenib. Could you please justify this time point? If I remember correctly sorafenib tends to have erratic absorption and may peak several hours after administration.
- Figure 3. May explicitly label x-axis “days after treatment”, as in Fig. 3A the days in x-axis are about days after tumor inoculation. It can be a bit confusing to the readers.
- Only A549 parental and transfected cell lines were used. How generalizable are the conclusions? May suggest to discuss a bit more about this.
- Please fix the fonts. Apparently a number of font types/font size/font bolding are used.
Author Response
In this manuscript titled “Absence of HTATIP2 Expression in A549 Lung Adenocarcinoma Cells Promotes Tumor Plasticity in Response to Hypoxic Stress”, the authors helped delineate the mechanism by which the absent expression of HTATIP2 contributes to tumor adaptation to hypoxia and the tumor aggressiveness. The report has successfully clarified more about the role of HTATIP as a tumor suppressor. In general, the data are robust and the manuscript is clearly written. The reviewer recommends acceptance of the manuscript for publication after addressing the minor issues below:
- Figure 2: Please indicates the P values of those * and ** in the figure legend.
Reply: We have added the following information to the legend to Figure 2 “*p < 0.05 and **p < 0.01 compared between vehicle and sorafenib treatment in the same cell line under the same culture condition, or between A549shNT and A549shHTATIP2 cell lines receiving the same treatment and under the same culture condition, or between normoxia and hypoxia conditions in the same cell line with the same treatment using the two-sample t-test.”.
- Line 196-198: Measurement of intratumoral oxygen saturation was performed at 2 hours after the last dose of sorafenib. Could you please justify this time point? If I remember correctly sorafenib tends to have erratic absorption and may peak several hours after administration.
Reply: Based on our previous experience, sorafenib has a tmax (time to reach the maximal concentration) value of about 4 hours in mice after oral administration. Therefore, the timing of a series of experiments that took place on the last day of the treatment was designed in such a way that the plasma and tumor samples could be collected about 4 hours after the last dose when the peak sorafenib concentrations were reached in plasma and tumor tissue. The PAI was performed 2 hours after dosing and took about 30 min to complete in each animal. After that, pimonidazole (the hypoxia marker) was administered to individual animals which were euthanized 90 minutes after the administration of pimondazole according to the manufacturer’s protocol. We were less concerned about whether the measurement of intratumoral oxygen saturation should be performed when the peak sorafenib concentration is reached. This is because, for any given drug, the steady state concentration should be reached after 5 half-lives when the drug is given repeatedly at a fixed dosage and dosing interval. Therefore, the steady-state concentration of sorafenib should be reached after 10 days of treatment. Since there is usually a time delay in the response-plasma concentration relationship due to the sluggishness of the response following the direct action of the drug at the target site, whether the peak sorafenib concentration is reached at a particular time should not have much impact on the overall intratumoral oxygen saturation, which is more likely to be associated with sorafenib’s antiangiogenic activity rather than its concentrations.
- Figure 3. May explicitly label x-axis “days after treatment”, as in Fig. 3A the days in x-axis are about days after tumor inoculation. It can be a bit confusing to the readers.
Reply: We have changed the x-axis label in Figure 3B to “Days after Start of Treatment”
- Only A549 parental and transfected cell lines were used. How generalizable are the conclusions? May suggest to discuss a bit more about this.
Reply: In the pilot study, which was briefly mentioned in the manuscript, we used 4 human lung cancer cell lines and found only A549 cell line was suitable for the study. Our data then showed that the in vivo study using the A549 xenograft model was more informative and more relevant than the in vitro investigations. Although it is unclear if the conclusion drawn from this study is generalizable, the approach developed in this study can be utilized to investigate the mechanism involved in the tumor response to hypoxic stress in other HTATIP2-deficient lung cancer cell lines. As per the reviewer’s suggestion, we have stated at the end of the conclusion section that “Given the emerging evidence of the association of low HTATIP2 protein expression with high risk of metastasis and poor prognosis in NSCLC , by elucidating the distinct role of HTATIP2 in orchestrating tumor adaptation to hypoxic stress, the present study sets the groundwork for further attempts to identify new potential targets for therapeutic intervention in HTATIP2-deficient NSCLC.” (Line 785-789)
- Please fix the fonts. Apparently a number of font types/font size/font bolding are used.
Reply: Changes in the font type and font size were probably due to the editing done by the journal editorial staff. The font type and font size have been fixed and are consistent with the journal format.
